# Nanobody-directed targeting of optogenetic tools to study signaling in the primary cilium

Jan N Hansen[1†], Fabian Kaiser[1†], Christina Klausen[1], Birthe Stüven[1], Raymond Chong[1], Wolfgang Bönigk[2], David U Mick[3], Andreas Möglich[4,5,6], Nathalie Jurisch-Yaksi[7,8], Florian I Schmidt[9,10]*, Dagmar Wachten[1,11]*

[1]Institute of Innate Immunity, Biophysical Imaging, Medical Faculty, University of Bonn, Bonn, Germany; [2]Department of Molecular Sensory Systems, Center of Advanced European Studies and Research (caesar), Bonn, Germany; [3]Center for Molecular Signaling (PZMS), Center of Human and Molecular Biology (ZHMB), Saarland University, School of Medicine, Homburg, Germany; [4]Lehrstuhl für Biochemie, Universität Bayreuth, Bayreuth, Germany; [5]Research Center for Bio-Macromolecules, Universität Bayreuth, Bayreuth, Germany; [6]Bayreuth Center for Biochemistry & Molecular Biology, Universität Bayreuth, Bayreuth, Germany; [7]Kavli Institute for Systems Neuroscience and Centre for Neural Computation, The Faculty of Medicine, Norwegian University of Science and Technology, Trondheim, Norway; [8]Department of Neurology and Clinical Neurophysiology, St. Olavs University Hospital, Trondheim, Norway; [9]Institute of Innate Immunity, Emmy Noether research group, Medical Faculty, University of Bonn, Bonn, Germany; [10]Core Facility Nanobodies, University of Bonn, Bonn, Germany; [11]Research Group Molecular Physiology, Center of Advanced European Studies and Research (caesar), Bonn, Germany

**\*For correspondence:**
fschmidt@uni-bonn.de (FIS);
dwachten@uni-bonn.de (DW)

[†]These authors contributed equally to this work

**Competing interests:** The authors declare that no competing interests exist.

**Abstract** Compartmentalization of cellular signaling forms the molecular basis of cellular behavior. The primary cilium constitutes a subcellular compartment that orchestrates signal transduction independent from the cell body. Ciliary dysfunction causes severe diseases, termed ciliopathies. Analyzing ciliary signaling has been challenging due to the lack of tools to investigate ciliary signaling. Here, we describe a nanobody-based targeting approach for optogenetic tools in mammalian cells and in vivo in zebrafish to specifically analyze ciliary signaling and function. Thereby, we overcome the loss of protein function observed after fusion to ciliary targeting sequences. We functionally localized modifiers of cAMP signaling, the photo-activated adenylyl cyclase bPAC and the light-activated phosphodiesterase LAPD, and the cAMP biosensor mlCNBD-FRET to the cilium. Using this approach, we studied the contribution of spatial cAMP signaling in controlling cilia length. Combining optogenetics with nanobody-based targeting will pave the way to the molecular understanding of ciliary function in health and disease.

## Introduction

Primary cilia are membrane protrusions that extend from the surface of almost all vertebrate cells. Primary cilia function as antennae that translate sensory information into a cellular response. The sensory function is governed by a subset of receptors and downstream signaling components that are specifically targeted to the cilium. This allows to orchestrate rapid signal transduction in a minuscule reaction volume, independent of the cell body. A central component of ciliary signaling is the second

messenger 3', 5'-cyclic adenosine monophosphate (cAMP) (*Johnson and Leroux, 2010*). The prime example is chemosensation in highly specialized olfactory cilia: odorant-induced activation of G-protein-coupled receptors (GPCRs) stimulates the synthesis of cAMP by the transmembrane adenylyl cyclase 3 (AC3). The ensuing increase in ciliary cAMP levels activates cyclic nucleotide-gated ion channels (CNG), resulting in a depolarization that spreads from the cilium to the synapse (*Kaupp, 2010*). In recent years, it emerged that cAMP also controls signaling in primary cilia. AC3 is highly enriched in primary cilia and widely used as a ciliary marker (*Antal et al., 2017*; *Bishop et al., 2007*). Loss-of-function mutations in the *ADCY3* gene, encoding for AC3, or loss of *ADCY3* expression cause monogenic severe obesity and increase the risk for type 2 diabetes (*Cao et al., 2016*; *Grarup et al., 2018*; *Nordman et al., 2005*; *Saeed et al., 2018*; *Siljee et al., 2018*; *Wang et al., 2009*). This has been attributed to the loss of AC3 function in neuronal primary cilia (*Siljee et al., 2018*; *Barroso, 2018*). Furthermore, the most prominent primary cilia signaling pathway, the Sonic hedgehog (Shh) pathway, utilizes cAMP as a second messenger in the cilium to transduce stimulation by Shh into a change in gene expression (*Moore et al., 2016*; *Mukhopadhyay et al., 2013*). Finally, the dynamic modulation of primary cilia length seems to be controlled by cAMP (*Besschetnova et al., 2010*; *Porpora et al., 2018*; *Jin et al., 2014*). However, as of yet, it has been impossible to manipulate cAMP dynamics in primary cilia independently from the cell body. Hence, the molecular details and dynamics of cAMP-signaling pathways in primary cilia remain largely unknown.

Optogenetics might be the key to overcome this issue, not least because it has proven to be a powerful method to manipulate and monitor cAMP dynamics in mouse sperm flagella, a specialized motile cilium (*Balbach et al., 2018*; *Mukherjee et al., 2016*; *Jansen et al., 2015*). The photo-activated adenylyl cyclase bPAC (*Stierl et al., 2011*) has been employed to increase flagellar cAMP levels by blue light (*Jansen et al., 2015*), and the FRET-based cAMP biosensor mlCNBD-FRET has been used to monitor cAMP dynamics in sperm flagella (*Mukherjee et al., 2016*). This cAMP tool kit has been complemented with the red light-activated phosphodiesterase LAPD that allows to decrease cAMP levels in a light-dependent manner (*Gasser et al., 2014*; *Stabel et al., 2019*). For primary cilia, the challenge is to specifically target these tools to the cilium to investigate cAMP signaling independent from the cell body. Free diffusion of proteins into the primary cilium is limited by the transition zone (TZ) at the base of the cilium (*Reiter et al., 2012*). Protein transport into and out of the cilium relies on the intraflagellar transport (IFT) machinery in combination with the BBSome, a multi-protein complex at the ciliary base (*Berbari et al., 2008*; *Loktev and Jackson, 2013*; *Nachury, 2018*; *Rosenbaum and Witman, 2002*). The combined action of IFT, BBSome, and TZ shape the unique ciliary protein composition (*Nachury and Mick, 2019*). To localize a given optogenetic tool to the primary cilium, the ciliary transport machinery needs to be hijacked. Common strategies involve direct fusion to the C terminus of either a full-length GPCR, for example the somatostatin receptor 3 (Sstr3) (*Berbari et al., 2008*; *Guo et al., 2019*), the 5-HT$_6$ receptor (*Moore et al., 2016*), to a ciliary protein, for example Arl13b (*Jiang et al., 2019*), or a truncated ciliary protein, for example the first 201 amino acids (aa) of the ciliary mouse Nphp3 (nephrocystin 3) protein (*Wright et al., 2011*; *Mick et al., 2015*). Fusion of Sstr3 to the N terminus of bPAC or mlCNBD-FRET has already been applied (*Mukherjee et al., 2016*; *Guo et al., 2019*). However, using a full-length GPCR as a fusion partner for ciliary localization might increase the abundance of this specific receptor in the cilium and, thereby, distort the ciliary set of signal receptors. This is particularly disadvantageous when analyzing ciliary cAMP signaling because fusion to GPCRs, which couple to cAMP signaling, might alter basal cAMP levels due to constitutive activity, as has been shown for the commonly used 5-HT$_6$ receptor (*Kohen et al., 2001*). In contrast, the truncated version of mNphp3 is sufficient to convey ciliary targeting (*Wright et al., 2011*), but lacks any other known functionality, whereby its fusion for ciliary targeting represents by large a less invasive targeting approach than using a complete GPCR. Thus, to target optogenetic tools to the primary cilium, we fused aa 1–201 of mNphp3 to their N terminus. To our surprise, this approach largely failed for LAPD and mlCNBD-FRET, because N-terminal fusion disrupts their optogenetic and biosensoric function, respectively. Therefore, we developed a new approach to hijack the ciliary transport machinery to target intracellular nanobodies. The nanobody-based approach allows to specifically target proteins to the primary cilium without directly fusing them to a ciliary protein. Instead, the nanobody is cilia-targeted and tows the proteins of interest to the cilium by binding to a tag contained in the protein of interest.

We show that the cilia-targeted nanobodies bind to eGFP or mCherry-containing proteins of interest in the cytosol, such as bPAC, LAPD, or mlCNBD-FRET. As a complex, the proteins then translocate into the primary cilium. Nanobody binding does not substantially impair protein function, that is light-dependent activation of bPAC and LAPD, or cAMP-induced conformational changes of mlCNBD-FRET. We demonstrate the validity of this approach both in vitro and in vivo in mammalian cells and zebrafish, respectively. Moreover, using nanobody-based ciliary targeting of bPAC, we study the role of ciliary versus cytosolic (cell body) cAMP signaling in controlling cilia length. Our approach principally extends to the ciliary targeting of any protein of interest, which is recognized by a nanobody. Thereby, the huge variety of genetically encoded tools to manipulate cellular signaling, for example membrane potential, protein-protein interactions, or enzymatic activities, and to monitor cellular signaling, for example dynamics of $Ca^{2+}$, pH, or the membrane potential, could be enriched in the primary cilium using our nanobody-based targeting approach. This strategy opens up new avenues for cilia biology and allows to address long-standing questions in cell biology.

## Results

### N-terminal fusion of optogenetic tools interferes with photoactivation

To utilize the optogenetic tools bPAC- or LAPD-mCherry in a cilium-specific manner, we first tested whether N-terminal fusion of mNphp3(201) is sufficient for targeting to the primary cilium. Indeed, fusions of both bPAC and LAPD predominantly localized to the primary cilium (*Figure 1A,B*). To test whether protein fusion interferes with the light-dependent activation of LAPD or bPAC, we measured LAPD or bPAC activity using $Ca^{2+}$ imaging. To this end, we used HEK293 cells stably expressing a cyclic nucleotide-gated (CNG) ion channel CNGA2-TM (HEK-TM) that conducts $Ca^{2+}$ upon cAMP binding (*Wachten et al., 2006*). HEK293 cells were not ciliated to directly compare the non-fused and the fused optogenetic tool. Activation of bPAC with a 465 nm light pulse increases intracellular cAMP levels, leading to a $Ca^{2+}$ influx, which is quantified using the fluorescence of a $Ca^{2+}$ indicator dye (*Figure 1C*). To measure LAPD activity, HEK-TM cells were pre-stimulated with NKH477, a water-soluble forskolin analog that activates transmembrane adenylyl cyclases (tmACs) and, thereby, increases cAMP levels, leading to a $Ca^{2+}$ influx. NKH477 stimulation was performed under illumination with 850 nm that deactivates LAPD, as previously described (*Gasser et al., 2014*; *Stabel et al., 2019*). When the $Ca^{2+}$ influx reached a steady-state, LAPD was activated by 690 nm light, decreasing cAMP levels and, thereby, the intracellular $Ca^{2+}$ concentration (*Figure 1C*). We measured the mNphp3(201)-bPAC-mCherry or mNphp3(201)-LAPD-mCherry activity and compared it to the non-ciliary tagged bPAC- or LAPD-mCherry proteins. Light stimulation of mNphp3(201)-bPAC-mCherry or bPAC-mCherry expressing HEK-TM cells resulted in a transient $Ca^{2+}$ increase, which was absent in mCherry-expressing control cells (*Figure 1D*). Repetitive light-stimulation with different light pulses reliably increased the intracellular $Ca^{2+}$ concentration in mNphp3(201)-bPAC-mCherry or bPAC-mCherry expressing HEK-TM cells (*Figure 1D*). Normalized peak amplitudes of the $Ca^{2+}$ signal evoked after the first light pulse were lower in mNphp3(201)-bPAC-mCherry than in bPAC-mCherry expressing HEK-TM cells (*Figure 1E*), indicating that the N-terminal fusion to a ciliary targeting sequence interferes with the light-dependent activation of bPAC. Next, responses of HEK-TM cells stably expressing mNphp3(201)-LAPD-mCherry or LAPD-mCherry to NKH477 stimulation were quantified: in both, mNphp3(201)-LAPD-mCherry and LAPD-mCherry expressing HEK-TM cells, NKH477 stimulation induced a $Ca^{2+}$ increase (*Figure 1F*). Activating LAPD with 690 nm light significantly decreased the intracellular $Ca^{2+}$ concentration in LAPD-mCherry expressing, but not in mNphp3(201)-LAPD-mCherry expressing HEK-TM cells (*Figure 1G*), demonstrating that N-terminal fusion to a ciliary targeting sequence interferes with the light-dependent activation of LAPD. Taken together, our results obtained with two different optogenetic tools revealed that fusion with the minimal ciliary targeting motif mNphp3(201) interfered with their light-dependent activation, thus hampering a direct targeting strategy that does not rely on introducing a functional GPCR to the cilium.

### Targeting optogenetic tools to the primary cilium using nanobodies

We next devised a combinatorial strategy that allows targeting to the primary cilium, while entirely avoiding N-terminal fusion. Rather, we fused our optogenetic tools with a fluorescent reporter (e.g.

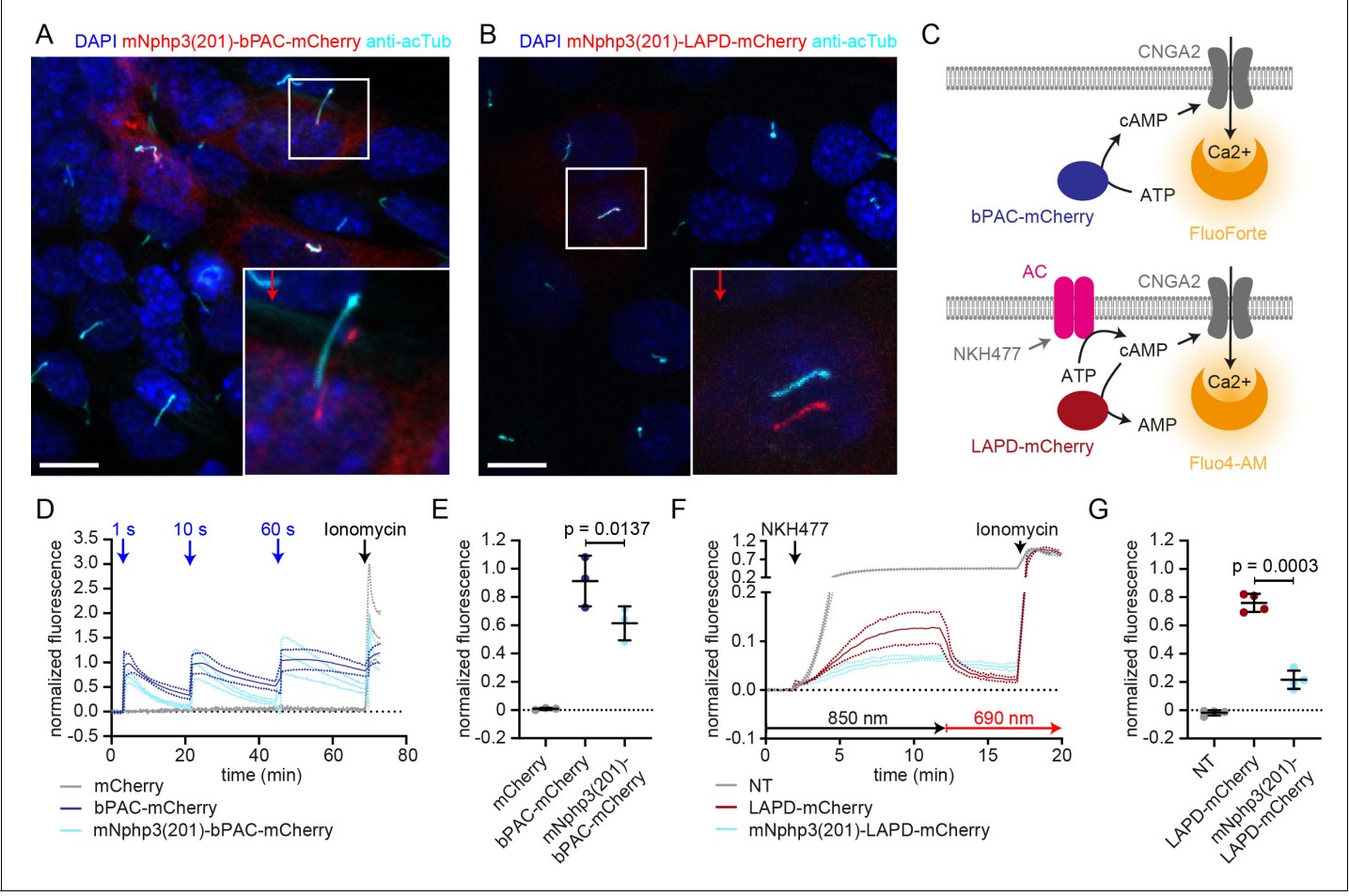

**Figure 1.** Direct ciliary targeting of optogenetic tools impairs protein function. (**A**) Localization of mNphp3(201)-bPAC-mCherry to primary cilia. mIMCD-3 cells expressing mNphp3(201)-bPAC-mCherry were labeled with an anti-acetylated tubulin antibody (cyan, ciliary marker) and with DAPI (blue) to label the DNA. The box indicates the position of the magnified view shown at the bottom right. Red arrow indicates the direction and the length of the shift of the respective fluorescence channel. Scale bar: 10 μm. (**B**) Localization of mNphp3(201)-LAPD-mCherry to primary cilia. mIMCD-3 cells expressing mNphp3(201)- LAPD-mCherry were labeled with an anti-acetylated tubulin antibody (cyan, ciliary marker) and DAPI (blue) to label the DNA. The box indicates the position of the magnified view shown at the bottom right. Red arrow indicates the direction and the length of the shift of the respective red channel. Scale bar: 10 μm. (**C**) Assays to measure bPAC or LAPD activity using $Ca^{2+}$ imaging. HEK293 cells express the CNGA2-TM ion channel, which opens upon cAMP binding and conducts $Ca^{2+}$ (HEK-TM) (*Wachten et al., 2006*). Light-dependent activation of bPAC increases intracellular cAMP levels, leading to a $Ca^{2+}$ influx, which was quantified using a fluorescent $Ca^{2+}$ dye (GFP-certified FluoForte). To measure LAPD activity, HEK-TM cells were pre-stimulated with 100 μM NKH477 to activate transmembrane adenylyl cyclases (AC), thus increasing cAMP levels. $Ca^{2+}$ influx was detected by a $Ca^{2+}$ dye (Fluo4-AM). (**D**) Quantification of bPAC activity. GFP-certified-FluoForte-loaded HEK-TM cells expressing mCherry only (grey), bPAC-mCherry (blue), or mNphp3(201)-bPAC-mCherry (cyan) were stimulated with 465 nm light pulses (1 mW/cm$^2$) of different length and the increase in the intracellular $Ca^{2+}$ concentration was measured. To evoke a maximal $Ca^{2+}$ response, cells were stimulated with 2 μM ionomycin. Data are shown as mean ± SD (dotted lines) for the normalized fluorescence (F-F(baseline))/(F(ionomycin)-F(baseline))/fraction of mCherry-positive cells, n = 3 independent experiments (each data point represents the average of a duplicate or triplicate measurement). (**E**) Mean peak amplitudes of the $Ca^{2+}$ signal at 3–6 min after the first light pulse. Data are shown as individual data points and mean ± SD, n = 3. (**F**) Quantification of LAPD activity. Fluo4-AM-loaded HEK-TM cells expressing LAPD-mCherry (red) or mNphp3(201)-LAPD-mCherry (cyan) were incubated with 100 μM NKH477 during continuous 850 nm light stimulation (0.5 μW/cm$^2$). At steady-state, light stimulation was switched to 690 nm (0.5 μW/cm$^2$). NT: non-transfected cells (grey). Data are shown as mean ± SD (dotted lines) for the normalized fluorescence (F-F(baseline))/(F(ionomycin)-F(baseline)). (**G**) Mean decrease of the $Ca^{2+}$ signal after 690 nm light stimulation (fraction of maximum value after NKH477 increase), determined over 45 s at 3 min after switching to 690 nm. Data are shown as individual data points and mean ± SD, n = 4 independent experiments (each data point represents the average of a duplicate or triplicate measurement); p-values calculated using a paired, two-tailed t-test are indicated. NT: non-transfected cells.

mCherry) at their C termini, which leaves photoactivation unaffected (*Jansen et al., 2015*; *Stabel et al., 2019*). To direct these proteins to primary cilia, we co-expressed a nanobody, which is directed against the tag (mCherry) and is fused to the ciliary targeting sequence mNphp3(201) at its N terminus. We hypothesize that the nanobody binds to its target in the cytoplasm, the nanobody-protein-complex is recognized by the ciliary targeting machinery and is then transported into the primary cilium (*Figure 2A*). We first tested the anti-mCherry nanobody VHH$_{LaM-2}$ (*Ariotti et al., 2018*; *Fridy et al., 2014*), fused to eGFP at the C terminus and mNphp3(201) at the N terminus, in mIMCD-3 cells. Indeed, the nanobody localized to primary cilia (*Figure 2B*). Next, we assessed whether nanobody binding was sufficient to traffic our optogenetic tools to the cilium. Co-expression of the nanobody with LAPD-mCherry resulted in ciliary localization of both the nanobody fusion-construct and LAPD-mCherry (*Figure 2C*). In contrast, LAPD-mCherry remained exclusively cytosolic in the absence of the nanobody (*Figure 2D*). A second nanobody directed against mCherry, VHH$_{LaM-4}$, (*Ariotti et al., 2018*; *Fridy et al., 2014*) also localized to primary cilia and resulted in ciliary localization of LAPD (*Figure 2—figure supplement 1A,B*), while a nanobody directed against eGFP (*Kirchhofer et al., 2010*) did not mediate ciliary localization of LAPD-mCherry (*Figure 2—figure supplement 1C*). The nanobody-based targeting approach also succeeded in localizing bPAC-mCherry to the primary cilium (*Figure 2E*). Taken together, nanobody-based targeting of optogenetic toosl was efficient and specific. Hence, we assume that our approach is generally applicable to target proteins of interest to cilia.

## Nanobody binding does not interfere with photoactivation

To test whether nanobody binding interferes with the light-dependent activation of LAPD or bPAC, we first tested their activity in non-ciliated cells to directly compare bound and non-bound optogenetic tools and then verified the optimal experimental condition in ciliated mIMCD-3 cells. To measure the activity in HEK-TM cells, bPAC- or LAPD-Cherry were co-expressed with the cilia-targeted mCherry nanobody mNphp3(201)-VHH$_{LaM-2}$-eGFP. In the absence of the nanobody, bPAC- and LAPD-mCherry displayed a cytosolic distribution (*Figure 2—figure supplement 2A,B*). In the absence of primary cilia, the mNphp3(201)-VHH$_{LaM-2}$-eGFP nanobody formed clusters in HEK-TM cells (*Figure 2—figure supplement 2C*), while the VHH$_{LaM-2}$-eGFP nanobody did not (*Figure 2—figure supplement 2D*). Co-expression of bPAC- or LAPD-mCherry with the mNPHP3(201)-tagged nanobody resulted in cluster localization of either bPAC or LAPD, demonstrating that the nanobody interacts with the mCherry fusion-proteins in the cytoplasm (*Figure 2—figure supplement 2E,F*).

To test bPAC or LAPD function in the presence of the nanobody, we compared the light-dependent activation of bPAC or LAPD in the presence or absence of the mNPHP3(201)-tagged mCherry nanobodies VHH$_{LaM-2}$ and VHH$_{LaM-4}$ (fused to either eGFP or a hemagglutinin HA-tag, respectively) or in the presence of a ciliary protein that does not interact with either bPAC- or LAPD-mCherry (Sstr3-eGFP). Under each condition, photoactivation of bPAC or LAPD activity was retained, demonstrating that interaction with the nanobody did not interfere with protein function (*Figure 2—figure supplement 3A,B*).

To scrutinize bPAC and LAPD activity during nanobody-binding in a ciliary context, we aimed to use cAMP biosensors, which can be targeted to the primary cilium. However, all reported biosensors spectrally overlap with the LAPD activation spectrum to some extend (*Klausen et al., 2019*). For bPAC, one biosensor is well suited to simultaneously activate bPAC and measure changes in cAMP levels: the red-shifted cAMP biosensor R-FlincA (*Ohta et al., 2018*). We co-expressed bPAC-eGFP and R-FlincA and first tested this approach in the cell body (*Figure 3A*). Photoactivation of bPAC-eGFP in HEK293 cells transiently increased the R-FlincA fluorescence, whereas a non-binding mutant sensor did not respond to bPAC photoactivation (*Figure 3B,C*), demonstrating that a light-stimulated increase in the intracellular cAMP concentration can be concomitantly measured using R-FlincA.

However, we failed to apply this orthogonal system to the primary cilium due to a low signal-to-noise ratio in the cilium. Therefore, we used an alternative approach to measure photoactivation of nanobody-targeted bPAC in the cilium. Mouse IMCD-3 cells were transfected with bPAC-mCherry and mNphp3(201)-VHH$_{LaM-2}$-HA to localize bPAC to the cilium. In addition, cells were transduced with the 5-HT$_6$-cADDis green cAMP biosensor, which is targeted to the primary cilium and reports an increase in cAMP with a decrease in cpEGFP fluorescence (*Moore et al., 2016*). We first verified 5-HT$_6$-cADDis sensor function by pharmacologically increasing cAMP levels (*Figure 3D–F*). To this

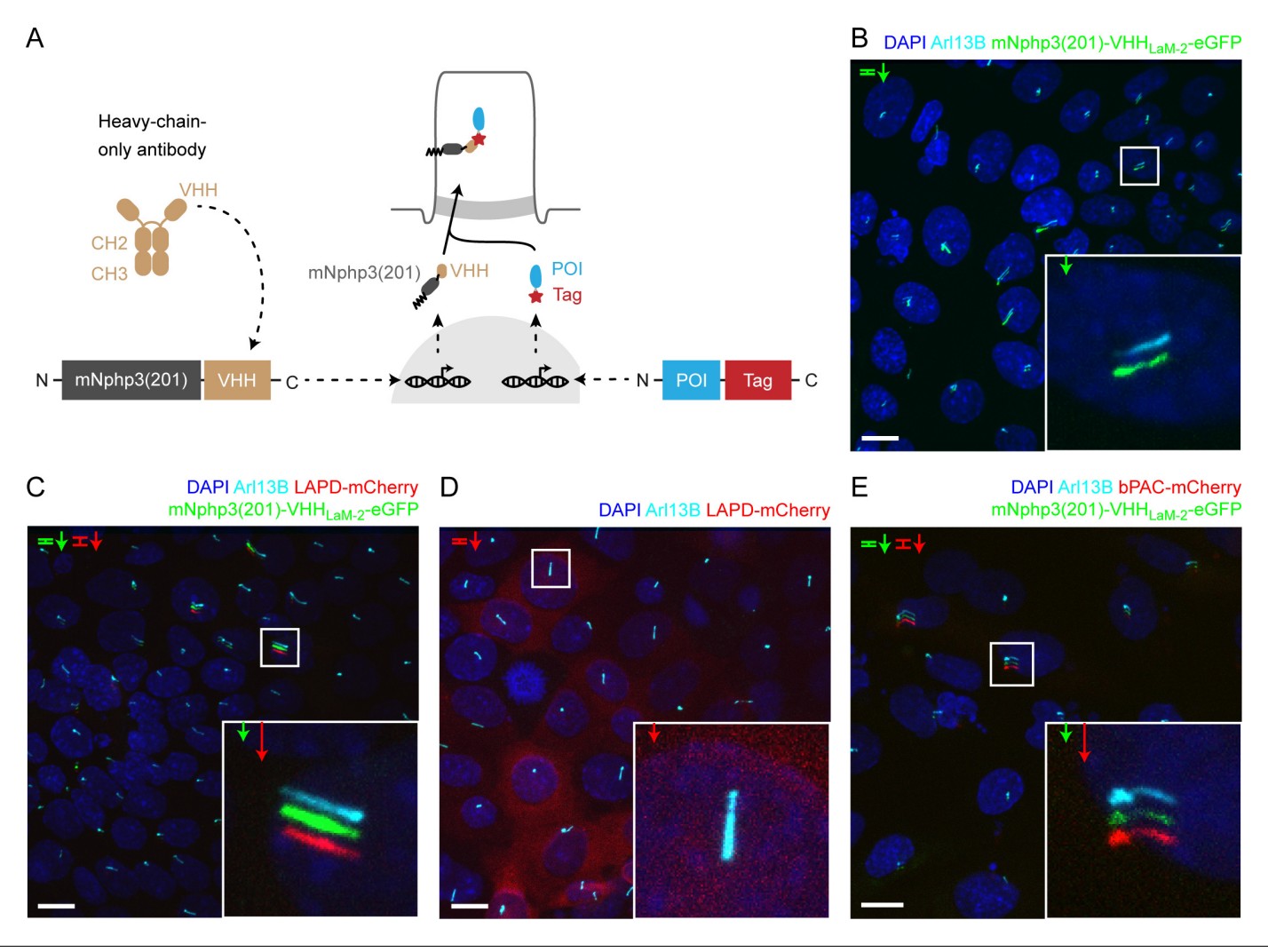

**Figure 2.** Targeting optogenetic tools to the primary cilium using nanobodies. (A) Schematic overview of the targeting approach. Nanobodies were fused to the C terminus of mNphp3(201) for ciliary localization. The protein of interest (POI) is co-expressed with a C-terminal tag or fusion partner that is recognized by the nanobody. Binding of the nanobody to the tag is expected to result in ciliary localization of the POI. (B) Localization of the anti-mCherry nanobody (VHH$_{LaM-2}$) to primary cilia. mIMCD-3 cells were transfected with mNphp3(201)-VHH$_{LaM-2}$-eGFP (green). (C) Localization of the anti-mCherry nanobody and LAPD-mCherry to primary cilia. mIMCD-3 cells were co-transfected with mNphp3(201)-VHH$_{LaM-2}$-eGFP (green) and LAPD-mCherry (red). (D) Cytoplasmic localization of LAPD-mCherry. mIMCD-3 cells were transfected with LAPD-mCherry (red). (E) Localization of the anti-mCherry nanobody and bPAC-mCherry to primary cilia. mIMCD-3 cells were co-transfected with mNphp3(201)-VHH$_{LaM-2}$-eGFP (green) and bPAC-mCherry (red). All cells shown in B-E were labeled with an Arl13B antibody (cyan, ciliary marker) and DAPI (blue). All scale bars: 10 μm. Boxes indicate the position of the magnified view shown at the bottom right. Arrows in different colors indicate the direction and the length of the shift of the respective fluorescence channel.

The online version of this article includes the following figure supplement(s) for figure 2:

**Figure supplement 1.** Nanobody-based ciliary targeting.

**Figure supplement 2.** Subcelluar localization of nanobody-targeted optogenetic tools.

**Figure supplement 3.** Activity measurements in HEK-TM cells.

end, we used 40 μM Forskolin, an activator of transmembrane adenylyl cyclases, or 250 μM IBMX, a broad-band phosphodiesterase inhibitor. Of note, pharmacological stimulation increases cAMP levels in the whole cell and not specifically in the primary cilium. The sensor reliably reported an increase in cAMP levels in the primary cilium evoked by either of the two stimuli (*Figure 3D,F*). Next, we analyzed cAMP levels in primary cilia in the presence of mNphp3(201)-VHH$_{LaM-2}$-HA and bPAC-mCherry or mCherry as control (*Figure 3G–I*). In this approach, however, the tools are not

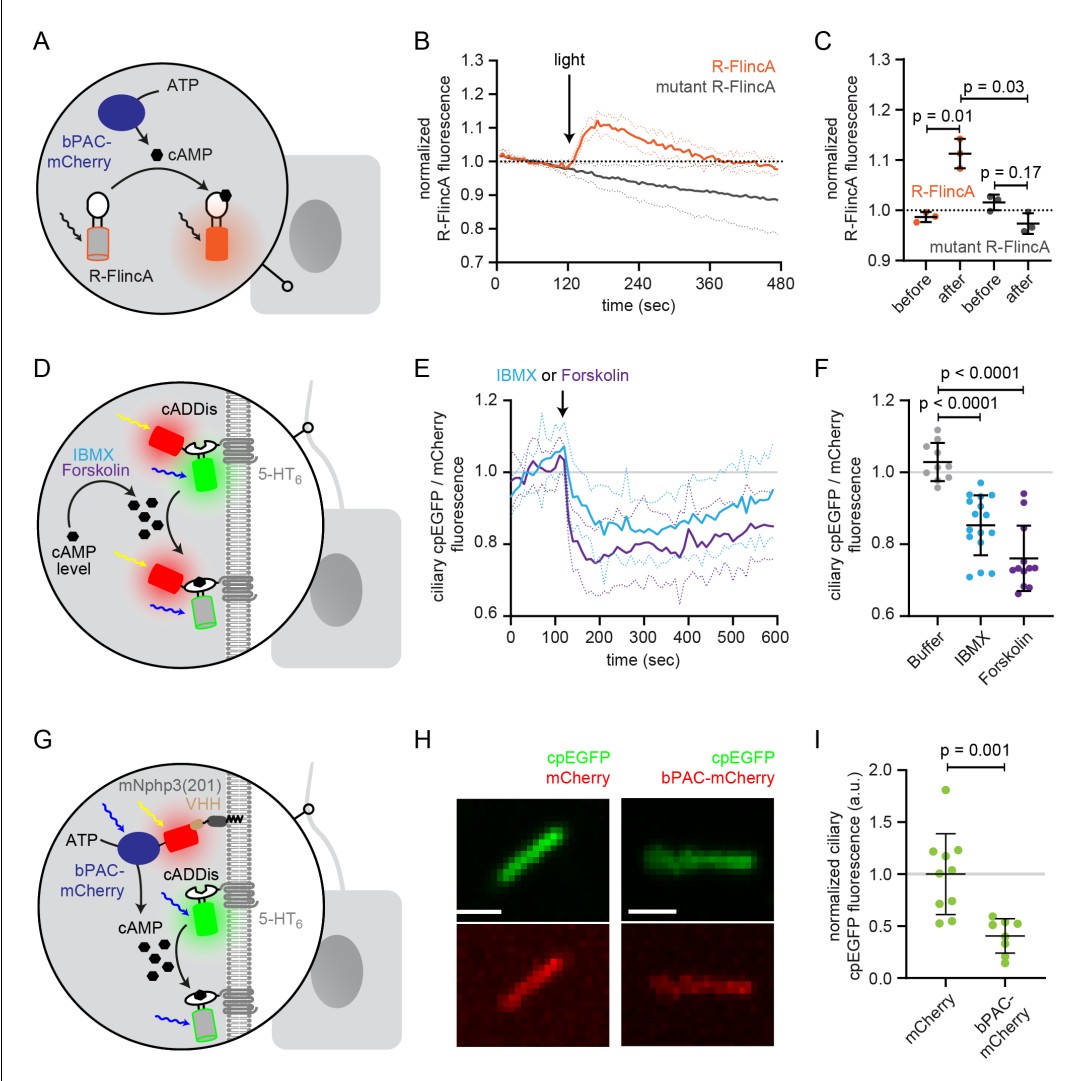

**Figure 3.** Functional characterization of bPAC in the cell body and cilium. (**A**) Schematic overview of the bPAC activity assay in non-ciliated HEK293 cells using R-FlincA (see **B-C**). (**B**) HEK293 cells were transfected with bPAC-eGFP and R-FlincA or the non-binding R-FlincA mutant (***Ohta et al., 2018***). The change in R-FlincA fluorescence was measured over time before and after photoactivation of bPAC (5 s, white light, 2.1 mW/cm$^2$ at 480 nm). Data are shown as mean (solid lines)± S.D. (dotted lines), n = 3 with 4 cells per experiment. (**C**) Normalized R-FlincA or R-FlincA mutant fluorescence directly before and for the maximal amplitude after photoactivation. Data extracted from C; p-values have been calculated using a paired, two-sided Student's t-test. (**D**) Schematic overview of the assay to measure ciliary cAMP dynamics using 5-HT$_6$-mCherry-cADDis after pharmacologically increasing cAMP levels (see E-F). (**E**) Ciliary cAMP dynamics measured using 5-HT$_6$-mCherry-cADDis. Cells were stimulated with 250 µM IBMX (light blue) or 40 µM Forskolin (purple). The normalized ratio of ciliary mCherry/cpEGFP fluorescence is shown as mean (solid lines)± S.D. (dotted lines); p-values have been calculated by paired, two-sided Student's t-test. (**F**) Mean change in the normalized ratio of ciliary mCherry/cpEGFP fluorescence 60–120 s after stimulation with buffer, IBMX, or Forskolin. Data are shown as individual data points, the mean ± S.D. is indicated; p-values have been calculated by a two-sided Mann-Whitney test. (**G**) Schematic overview of the assay to measure light-evoked ciliary cAMP dynamics after bPAC stimulation using 5-HT$_6$-cADDis (see H-I). (**H**) 5-HT6-cADDis fluorescence in cilia with mNphp3(201)-VHH$_{LaM-2}$-HA targeted mCherry or bPAC-mCherry in the first frame of imaging. Scale bar: 2 µm. (**I**) Mean normalized ciliary cpEGFP fluorescence in the first frame. All data have been normalized to the mean cpEGFP fluorescence in the mCherry control. Data are shown as individual data points, the mean ± S.D. is indicated; p-values have been calculated by unpaired, two-sided Student's t-test.

spectrally separated as both bPAC and the cAMP biosensor are activated/excited by blue light (here: 488 nm). Thus, measuring the cpEGP fluorescence of the cADDis sensor concomitantly activates bPAC. Indeed, as soon as the measurement was started, bPAC was activated, cAMP levels increased and, in turn, the cpEGFP fluorescence of the cADDis sensor was significantly lower compared to the cpEGFP fluorescence in cilia that contained mCherry only (***Figure 3H,I***), demonstrating

that photoactivation of bPAC increased cAMP levels in the cilium. Thus, even though the approach lacks the spectral separation, it confirms light-induced changes of cAMP levels in the cilium and demonstrates that nanobody-based targeting of bPAC-mCherry to the cilium increases ciliary cAMP levels after photoactivation.

In summary, our nanobody-based approach provides a versatile means for ciliary targeting without interfering with protein function.

## Targeting of a genetically-encoded biosensor to the primary cilium

We previously engineered and applied a genetically encoded biosensor, named mlCNBD-FRET, to measure cAMP dynamics in motile cilia (*Mukherjee et al., 2016*). We already demonstrated targeting of this sensor to primary cilia by fusing it to the C terminus of Sstr3 (*Mukherjee et al., 2016*). However, Sstr3 is a functional GPCR, which may interfere with ciliary signaling, in particular cAMP signaling, upon overexpression. We aimed to optimize the targeting approach by fusing mNphp3 (201) to the N terminus of mlCNBD-FRET. While the biosensor localized to the primary cilium (*Figure 4—figure supplement 1A*), biosensor function was severely impaired as mlCNBD-FRET no longer responded to changes in cAMP levels (*Figure 4—figure supplement 1B,C*). We thus tested whether the biosensor can be targeted to primary cilia using our nanobody-based approach without interfering with protein function. The mlCNBD-FRET sensor consists of the FRET pair cerulean and citrine (*Mukherjee et al., 2016*). Both fluorescent proteins are recognized by the nanobody VHH$_{enhancer}$ directed against eGFP (*Kirchhofer et al., 2010*; *Kubala et al., 2010*). Fusion of mNphp3(201) to the N terminus of the anti-eGFP nanobody also resulted in ciliary localization (*Figure 4A*). In the absence of the nanobody, mlCNBD-FRET was uniformly distributed throughout the cytosol, whereas co-expression with the mNphp3(201)-tagged nanobody resulted in ciliary localization of mlCNBD-FRET (*Figure 4B,C*). To test whether nanobody interaction impaired mlCNBD-FRET function, we performed FRET imaging in HEK293 cells expressing mlCNBD-FRET in the presence or absence of the nanobody. Similar to the anti-mCherry nanobody, the cilia-targeted eGFP nanobody mNphp3(201)-VHH$_{enhancer}$-mCherry showed a more clustered subcellular localization in HEK293 cells in the absence of primary cilia formation (*Figure 4—figure supplement 1D*). Consistently, when binding to the nanobody, mlCNBD-FRET also formed clusters within the cytosol (*Figure 4—figure supplement 1E*), which did not occur in the presence of mCherry only (*Figure 4—figure supplement 1F*). To functionally test the FRET sensor in the presence of the nanobody, we first assessed the impact of the nanobody on the fluorescence intensity of the two fluorophores, cerulean and citrine. HEK293 cells were transfected with cerulean or citrine and the eGFP VHH$_{enhancer}$-mCherry nanobody or mCherry only. The fluorescence intensity of cerulean or citrine was normalized to the mCherry fluorescence in the same cell. Both cerulean and citrine showed an increase in fluorescence in the presence of the nanobody compared to the mCherry control as previously described (*Kirchhofer et al., 2010*), but the relative change for each of the fluorophores was not substantially different (*Figure 4—figure supplement 1G*). To test whether mlCNBD-FRET:nanobody complexes still respond to changes in cAMP levels, we first measured cAMP-induced FRET changes in non-ciliated HEK293 cells and then in ciliated mIMCD-3 cells. To increase the intracellular cAMP concentration, cells were stimulated with 20 μM isoproterenol, which stimulates AC activity through signaling via GPCRs (G-protein-coupled receptors). We analyzed FRET changes in HEK293 mlCNBD-FRET cells co-expressing mNphp3(201)-VHH$_{enhancer}$-mCherry or the non-targeted VHH$_{enhancer}$-mCherry nanobody (*Figure 4D*). In the presence of the VHH$_{enhancer}$-mCherry nanobody, the FRET response to stimulation with isoproterenol remained unchanged (*Figure 4E,F*) and also interaction with the mNphp3 (201)-tagged nanobody only marginally reduced the FRET response and generally left the reporter functional (*Figure 4E,F*).

After having verified biosensor function in the presence of the nanobody in non-ciliated cells, we performed FRET imaging in cilia of mIMCD-3 cells co-expressing mlCNBD-FRET and mNphp3(201)-VHH$_{enhancer}$-mCherry (*Figure 4G*). In response to stimulation with 250 μM IBMX to increase cAMP levels, the ciliary-localized mlCNBD-FRET responded with a change in FRET, whereas buffer addition did not change FRET (*Figure 4H,I*, *Figure 4—figure supplement 1H*, *Video 1*), demonstrating that the nanobody-targeted mlCNBD-FRET sensor can be used to study cAMP dynamics in the primary cilium. In conclusion, the nanobody-based approach applies not only for targeting optogenetic tools, but also genetically encoded biosensors to the primary cilium.

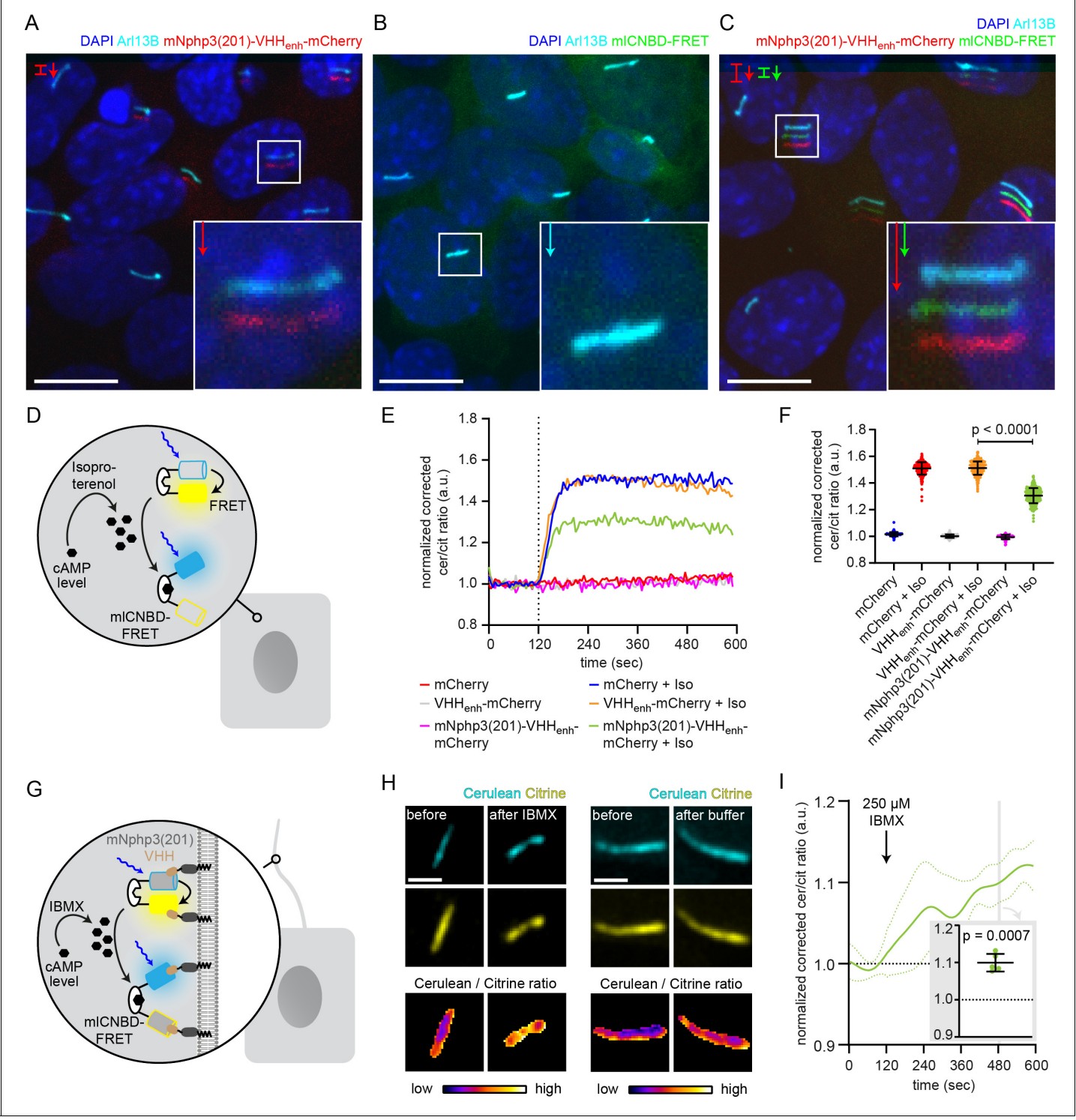

**Figure 4.** Functional characterization of nanobody-targeted cAMP biosensor. (**A**) Localization of the mNphp3(201)-VHH$_{enhancer}$-mCherry anti-eGFP nanobody to primary cilia. (**B./C**) Localization of mlCNBD-FRET in mIMCD-3 cells in the (**B**) absence or (**C**) presence of mNPHP3(201)-VHH$_{enhancer}$-mCherry. (**D**) Schematic overview of mlCNBD-FRET imaging in non-ciliated HEK293 cells (see E-F). (**E**) FRET imaging in HEK293 mlCNBD-FRET cells transiently co-expressing mCherry, VHH$_{enhancer}$-mCherry, or mNphp3(201)-VHH$_{enhancer}$-mCherry under control conditions or after stimulation with 20 µM isoproterenol (Iso, addition depicted with dotted line). Data are shown as mean (n = 3 independent experiments, 30–90 cells per experiment). (**F**) Comparison of maximal change for data shown in E. Data are presented as individual data points and mean ± S.D.; p-value calculated using an unpaired, two-tailed Mann-Whitney test is indicated. (**G**) Schematic overview of mlCNBD-FRET imaging in the primary cilium of mIMCD-3 cells (see H-I). (**H**) FRET imaging in primary cilia of mIMCD-3 cells expressing mlCNBD-FRET and mNphp3(201)-VHH$_{enhancer}$-mCherry. Cells have been stimulated with

*Figure 4 continued on next page*

Figure 4 continued

250 µm IBMX (left) or buffer only (right). Cerulean and citrine are shown before and after stimulation with IMBX. The change in cerulean/citrine ratio is shown below (color-scheme indicated at the bottom). Scale bar: 2 µm. (I) Time course of mean change in FRET (dark green line)± S.D. (dotted green line) for data set, exemplary shown in H; n = 5. Inset: each data point shows the time-average per cilium at the position indicated by grey box; one-sample Student's t-test compared to 1.0 indicated.

The online version of this article includes the following figure supplement(s) for figure 4:

**Figure supplement 1.** Characterization of the ciliary-targeted cAMP mlCNBD-FRET biosensor.

## Applying the nanobody-based ciliary targeting approach in vivo

Having shown that our bipartite strategy for localization to primary cilia works in vitro, we wondered whether we could also target proteins of interest to primary cilia in vivo. We first confirmed the ciliary localization of the nanobody in vivo by injecting mRNA of the anti-mCherry mNphp3(201)-VHH$_{LaM-2}$-eGFP nanobody into nacre (mitfa$^{-/-}$) zebrafish embryos, which are transparent and, therefore, widely used for fluorescence imaging (*Lister et al., 1999*). The cilia-targeted nanobody was expressed and localized to cilia in all tissues, including the developing neural tube, the primary and motile cilia of the spinal cord (*Kramer-Zucker et al., 2005*), and the eye (*Figure 5A,B,C*) and allowed to mark cilia in an in vivo imaging approach (*Figure 5—figure supplement 1A*). Localization of the nanobody to cilia is similar to the previously described bactin:arl13b-gfp transgenic line, where GFP is fused to the ciliary Arl13B protein (*Figure 5—figure supplement 1B*; *Borovina et al., 2010*; *Olstad et al., 2019*). To test whether the cilia-targeted nanobody can also direct proteins to primary cilia in vivo, we injected mRNA of the anti-mCherry nanobody fusion mNphp3(201)-VHH$_{LaM-2}$-eGFP into transgenic zebrafish embryos ubiquitously expressing RFP (Ubi:zebrabow) (*Pan et al., 2013*), which is also bound by the anti-mCherry nanobody. In the absence of the nanobody, RFP was distributed in the cytosol (*Figure 5D*, *Figure 5—figure supplement 1C*). In the presence of the nanobody, RFP was highly enriched in primary cilia (*Figure 5E*, *Figure 5—figure supplement 1D*), demonstrating that our nanobody-based approach efficiently targets ectopically expressed proteins to primary cilia and to motile cilia in vitro and in vivo.

## Investigating the spatial contribution of cAMP signaling to cilia length control

The primary cilium is a dynamic cellular structure that assembles and dissembles in accordance with the cell cycle (*Keeling et al., 2016*; *Kim and Tsiokas, 2011*; *Wang et al., 2019*). The interplay between assembly and disassembly determines the length of the primary cilium. cAMP-dependent signaling pathways have been shown to regulate cilia length (*Besschetnova et al., 2010*; *Porpora et al., 2018*; *Jin et al., 2014*; *Avasthi et al., 2012*; *Kwon et al., 2010*). Changes in cAMP signaling to study cilia length control have only been evoked using pharmacology, lacking spatial resolution and targeting both, the cilium and cell body. However, it is generally accepted that cAMP signaling occurs within defined subcellular compartments to evoke a specific cellular response (*Johnstone et al., 2018*). Whether an increase in cAMP levels in either the cilium or the cell body is sufficient to evoke a change in ciliary length, is not known. Thus, it is not surprising that it has been controversially discussed whether an increase in the intracellular cAMP concentration, evoked by pharmacological stimulation, results in an increase or decrease in cilia length, (*Besschetnova et al., 2010*; *Porpora et al., 2018*), as spatial cAMP signaling might evoke a differential response, which is impossible to reveal using pharmacology. We also performed pharmacological stimulation of cAMP synthesis in mIMCD-3 cells using Forskolin and analyzed the change in cilia length. To analyze cilia length in an automated and unbiased fashion in 3D, we

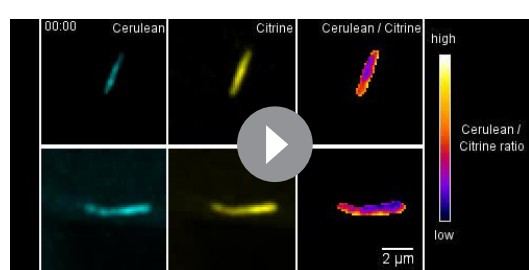

**Video 1.** FRET imaging in primary cilia. mIMCD-3 cells expressing mlCNBD-FRET and mNphp3(201)-VHH$_{enhancer}$-mCherry have been stimulated with 250 µm IBMX or buffer only.
https://elifesciences.org/articles/57907#video1

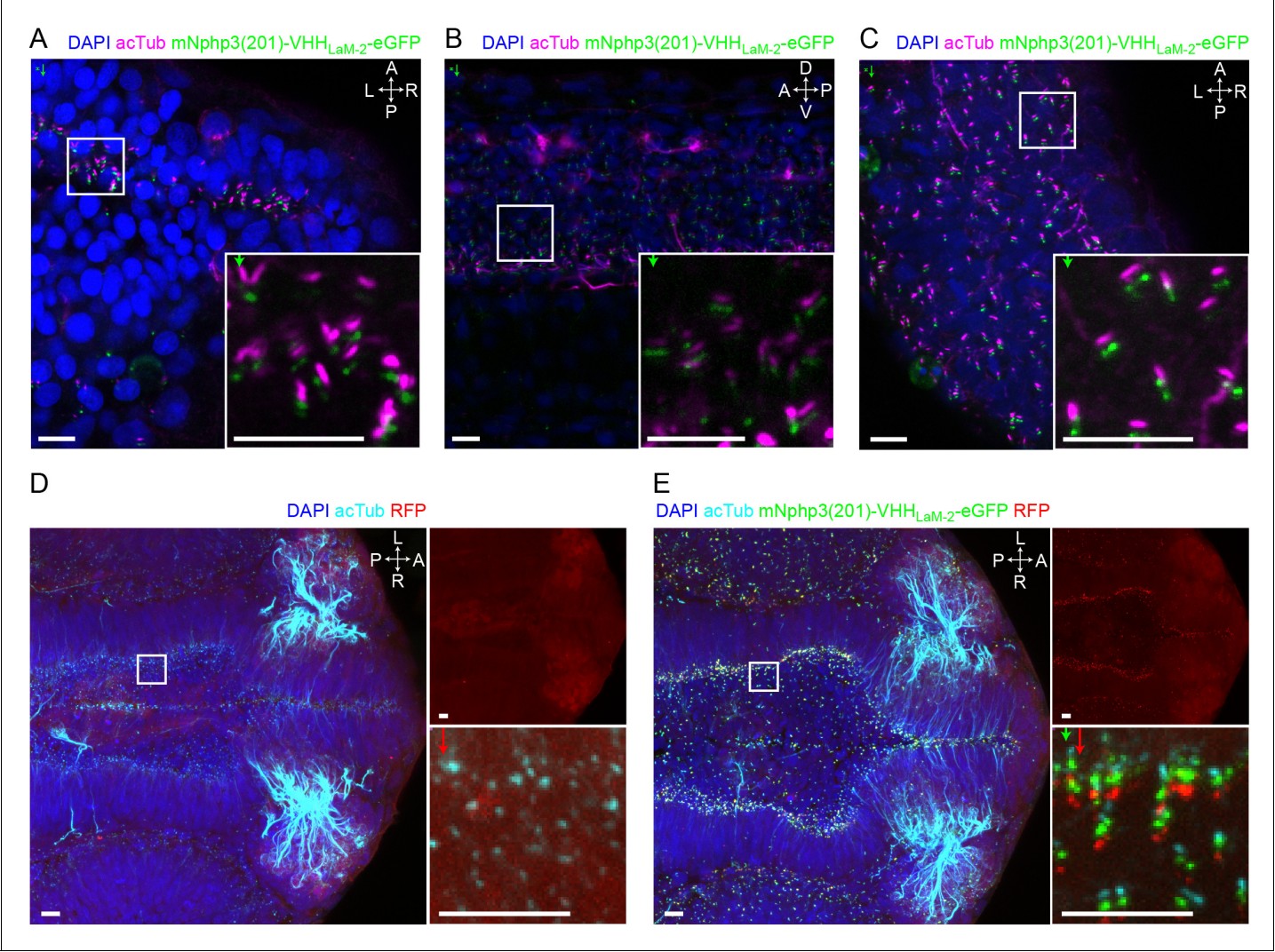

**Figure 5.** Nanobody-based ciliary protein targeting in vivo. (**A**) Nanobody localization in the neural tube of a zebrafish embryo. The mRNA of the anti-mCherry mNphp3(201)-VHH$_{LaM-2}$-eGFP nanobody was injected into nacre zebrafish embryos. Embryos were stained with an anti-acetylated tubulin antibody (magenta, ciliary marker), an anti-GFP antibody (green), and DAPI (blue). (**B**) See A. for spinal cord. (**C**) See A. for eye. (**D**) RFP (red) expression in the neural tube of Ubi:zebrabow (*Pan et al., 2013*) transgenic embryos. (**E**) RFP (red) expression in the neural tube of Ubi:zebrabow (*Pan et al., 2013*) transgenic embryos, injected with mRNA of the anti-mCherry mNphp3(201)-VHH$_{LaM-2}$-eGFP nanobody. Scale bars: 20 μm, magnified view: 10 μm. Boxes indicate the position of the magnified views shown at the bottom right as inset (**A-C**) or as a separate panel next to the overview image (**D, E**). Arrows in different colors indicate the direction and the length of the shift of the respective fluorescence channel. The upper right panel in D and E shows the RFP channel only, the bottom right panel shows the magnified view. A: anterior, P: posterior, L: left, R: right, D: dorsal, V: ventral. All images were taken from fixed samples.

The online version of this article includes the following figure supplement(s) for figure 5:

**Figure supplement 1.** Nanobody-based ciliary protein targeting in vivo.

developed an ImageJ plug-in called CiliaQ. In fact, the response was quite variable and we did not observe a significant change in cilia length (*Figure 6A*). Thus, we set out to investigate the spatial contribution of an increase in cAMP levels in the cilium or cell body in regulating cilia length.

To this end, we used a monoclonal IMCD-3 cell line stably expressing bPAC-mCherry in combination with the mNphp3(201)-VHH$_{Lam-2}$-eGFP nanobody. Since ectopic expression of a ciliary protein may result in an increase of the cilia length (*Guadiana et al., 2013*; *Koschinski and Zaccolo, 2017*), we first tested whether expression of the mNphp3(201)-tagged nanobody in the cilium had an impact on the length of the cilium. Indeed, ectopic expression of the mNphp3(201)-tagged nanobody resulted in longer cilia compared to non-transfected control cells. There was a linear

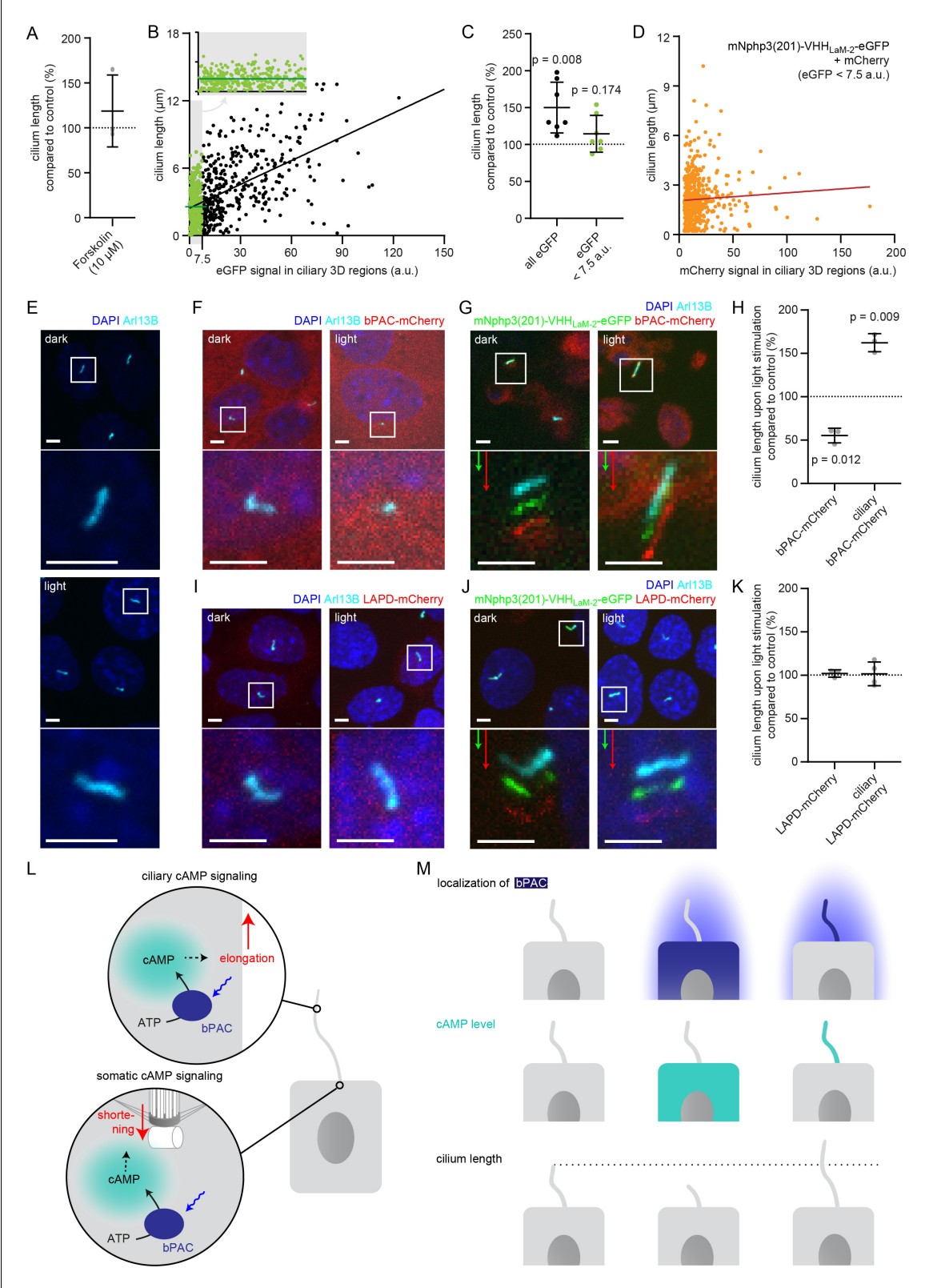

**Figure 6.** Controlling cilia length using optogenetics. (**A**) Cilia length of mIMCD-3 cells stimulated for 1 hr with 10 μM Forskolin (solvent: DMSO), normalized to the DMSO control. Data are shown as mean ± S.D., n = 3 with at least 40 cells per experiment. (**B**) Correlation of cilia length and eGFP fluorescence (a.u., average ciliary fluorescence of non-transfected control cells was subtracted) in the cilium in mIMCD-3 cells transiently expressing mNphp3(201)-VHHLaM-2-eGFP. Below 7.5 a.u., the cilia length is independent of the eGFP fluorescence (see inset, values are highlighted in green,

*Figure 6 continued on next page*

Figure 6 continued

slope not different from zero, correlation: p=0.07), whereas including values > 7.5 a.u., there is a linear correlation between the cilia length and the eGFP fluorescence in the cilium (slope different from zero, correlation: p<0.0001). (C) Length of cilia that show mNphp3(201)-VHH$_{LaM-2}$-eGFP localization and an eGFP fluorescence < 7.5 a.u., normalized to equally treated, non-transfected (NT) control cells. Data are shown as mean ± S.D., n = 7 with at least 18 cilia per experiment; p-values determined using unpaired, two-tailed Student's t-test are indicated. (D) Correlation of cilia length and mCherry fluorescence in the cilium in mIMCD-3 cells transiently expressing mNphp3(201)-VHHLaM-2-eGFP and mCherry. Only cilia with an eGFP fluorescence below 7.5 a.u. were taken into account. There is no linear correlation between the mCherry fluorescence and cilia length (slope not different from zero, correlation: p=0.2). (E) mIMCD-3 cells (non-transfected, NT) kept in the dark (top) or stimulated with light (bottom, 1 hr, 465 nm, 38.8 μW/cm$^2$) (F) mIMCD-3 bPAC-mCherry cells kept in the dark (left) or stimulated with light (right, 16 hr, 465 nm, 38.8 μW/cm$^2$). (G) mIMCD-3 bPAC-mCherry transiently transfected with mNphp3(201)-VHH$_{LaM-2}$-eGFP kept in the dark (left) or stimulated with light (right, 1 hr, 465 nm, 38.8 μW/cm$^2$). (H) Normalized cilia length after light stimulation (left 1 hr, right 16 hr; 465 nm, 38.8 μW/cm$^2$) for mIMCD-3 bPAC-mCherry cells with or without transiently expressing mNphp3(201)-VHH$_{LaM-2}$-eGFP. Only cilia with an eGFP fluorescence < 7.5 a.u. were included and each data point was normalized to control cells. Data are shown as mean ± S.D., n = 3 with at least 25 cells per experiment; p-values determined using one-sample Student's t-test compared to 100% are indicated. (I) mIMCD-3 LAPD-mCherry cells kept in the dark (left) or stimulated with light (right, 16 hr, 630 nm, 42.3 μW/cm$^2$). (J) mIMCD-3 LAPD-mCherry transiently transfected with mNphp3(201)-VHH$_{LaM-2}$-eGFP kept in the dark (left) or stimulated with light (right, 16 hr, 630 nm, 42.3 μW/cm$^2$). (K) Normalized cilia length after light stimulation (16 hr, 630 nm, 42.3 μW/cm$^2$) for mIMCD-3 LAPD-mCherry with or without transiently expressing mNphp3(201)-VHH$_{LaM-2}$-eGFP. Only cilia with an eGFP fluorescence < 7.5 a.u. were included and each data point was normalized to control cells. Data are shown as mean ± S.D., n = 3–4 with at least 18 cells per experiment; p-values determined using one-sample Student's t-test compared to 100% are indicated. Cells in E-G and I-J were stained with an Arl13B antibody (cyan) and DAPI (blue). All boxes indicate the magnified view below. Arrows indicate the direction and the length of the shift of the respective same-colored fluorescence channel. Scale bar for all images: 3 μm. (L) Spatial cAMP signaling controlling cilia length. Our data suggest a model, in which cAMP signaling in the cell body, stimulated by photoactivation of bPAC and an increase in cAMP levels, causes primary cilia shortening, whereas an increase of cAMP levels in the cilium results in primary cilia elongation. (M) Summary of the correlation between bPAC localization and photoactivation, cAMP levels, and cilia length.

The online version of this article includes the following figure supplement(s) for figure 6:

**Figure supplement 1.** cAMP levels and ciliary length in mIMCD-3 cells expressing LAPD.

correlation between the expression level of the mNphp3(201)-tagged nanobody in the cilium and cilia length: the higher the expression (determined by eGFP fluorescence), the longer the cilia (*Figure 6B*), as has been reported previously for ectopic expression of membrane proteins in the cilium (*Guadiana et al., 2013*). However, in the low expression regime, that is < 7.5 a.u. eGFP fluorescence, there was no linear correlation between the expression level of the mNphp3(201)-tagged nanobody in the cilium and cilia length (*Figure 6B* inset, *Figure 6C*), demonstrating that ectopic expression of the mNphp3(201)-tagged nanobody in the cilium at a low level does not change ciliary length. To verify whether the mNphp3(201)-tagged nanobody also does not alter cilia length in the low mNphp3(201)-VHH$_{Lam-2}$-eGFP expression regimes (<7.5 a.u.) while in complex with its target, we analyzed mIMCD-3 cells co-expressing mNphp3(201)-VHH$_{Lam-2}$-eGFP and mCherry. In the low mNphp3(201)-VHH$_{Lam-2}$-eGFP expression regimes (<7.5 a.u.), there was no linear correlation between the mCherry fluorescence and cilia length (*Figure 6D*), demonstrating that targeting the mNphp3(201)-tagged nanobody in complex with another protein to the cilium does not alter cilia length. Our results underline that the amount of protein in the cilium has to be carefully titrated and a thorough analysis is needed to rule out any unspecific effects caused by ectopic expression of ciliary proteins. In the following, we hence only analyzed cilia with a mNphp3(201)-VHH$_{Lam-2}$-eGFP expression level of < 7.5 a.u. We compared the change in cilia length upon photoactivation of bPAC-mCherry either in the cell body or in the presence of mNphp3(201)-VHH$_{Lam-2}$-eGFP in the cilium (*Figure 6E–H*). No light-dependent increase in ciliary length was observed in non-transfected cells (*Figure 6E*). Stimulating cAMP synthesis by light in the cell body significantly reduced cilia length (*Figure 6F,H*), whereas stimulating cAMP synthesis in the cilium significantly increased cilia length (*Figure 6G,H*).

Next, we complemented our analysis by investigating whether reducing cAMP levels in either the cell body or the cilium using photoactivation of LAPD also changed cilia length. We compared the change in cilia length upon photoactivation of LAPD-mCherry either in the cell body or, in the presence of mNphp3(201)-VHH$_{LaM-2}$-eGFP, in the cilium. However, neither stimulation of LAPD activity in the cell body nor in the cilium had an effect on cilia length (*Figure 6I–K*), although photoactivation of LAPD reduced basal cAMP levels (*Figure 6—figure supplement 1A*). Thus, the cAMP-dependent signaling pathways that control the length of the cilium seem to be sensitive to an increase, but not to a decrease of the basal cAMP concentration. In summary, cAMP-dependent signaling pathways in

the cell body or the cilium evoke opposing effects on cilia length control, demonstrating how compartmentalized cAMP signaling determines specific ciliary and, thereby, cellular functions (*Figure 6L, M*).

## Discussion

The primary cilium constitutes a unique subcellular compartment. However, our understanding of how specific ciliary signaling pathways control cellular functions is still limited. Optogenetics is an apt method to measure and manipulate ciliary signaling independent of the rest of the cell. We have developed a nanobody-based approach to target optogenetic tools or genetically-encoded biosensors to primary cilia in vitro and in vivo, while preserving protein function. This novel strategy principally extends to other proteins of interest and subcellular domains. The only requirement is fusion of the protein of interest to a tag or partner that is recognized by the nanobody, without impairing protein function. This is combined with fusion of the nanobody to a targeting sequence for the specific subcellular compartment. This approach is particularly useful for proteins, whose function is impaired by fusion to a targeting sequence, as we have shown here for the light-activated phosphodiesterase LAPD and the mlCNBD-FRET sensor.

Optogenetic tools have been used in motile cilia, that is sperm flagella (*Balbach et al., 2018*; *Jansen et al., 2015*), where no specific targeting sequence is needed. When expressed under the control of the sperm-specific protamine-1 promoter, bPAC exclusively localized to sperm flagella and allowed to control sperm signaling and function by light (*Jansen et al., 2015*). However, this approach does not work for primary cilia. Genetically-encoded biosensors for cAMP and $Ca^{2+}$ have already been targeted to primary cilia (*Moore et al., 2016*; *Mukherjee et al., 2016*; *Jiang et al., 2019*; *Delling et al., 2013*; *Delling et al., 2016*). These sensors have been fused to the C terminus of full-length GPCRs, for example 5-HT6 (5-hydroxytryptamine receptor 6) (*Moore et al., 2016*). Yet, overexpression of ciliary proteins like 5-HT6 or Arl13b caused abnormal cilia growth (*Jiang et al., 2019*; *Guadiana et al., 2013*), limiting this targeting approach. Our results demonstrate that also high expression levels of the nanobody in the cilium increased cilia length (*Figure 6B*). Thus, the amount of protein expressed in the cilium has to be carefully titrated and a thorough analysis is needed to rule out any unspecific effects caused by ectopic expression of the protein and reveal the specific contribution of a protein of interest to ciliary signaling and function.

Targeting proteins to a specific location using nanobodies has been applied to study protein-protein interaction in cells (*Herce et al., 2013*) and *Drosophila* (*Harmansa et al., 2017*). Herce et al. fused an anti-GFP nanobody to the lac repressor to localize the nanobody and its GFP-fused interaction partner to the nucleus. Using this approach, the authors have studied binding and disruption of p53 and HDM2 (human double minute 2), one of the most important protein interactions in cancer research (*Herce et al., 2013*). Harmansa et al. have mislocalized transmembrane proteins, cytosolic proteins, and morphogens in *Drosophila* to study the role of correct protein localization for development in vivo (*Harmansa et al., 2017*). Targeting to organelles using nanobodies has also been achieved for two specific proteins, p53 and survivin (*Beghein et al., 2016*; *Steels et al., 2018*). For primary cilia, recombinantly expressed fusion proteins fused to nanobodies have been used to investigate the ciliary diffusion barrier by diffusion-to-capture assays (*Breslow et al., 2013*) and determined a free diffusion across the ciliary barrier only for proteins < 100 kDa (*Kee et al., 2012*). Although larger proteins can enter cilia by active transport processes, this size cutoff may limit the size of proteins that can be targeted to cilia by a piggyback mechanism using a cilia-localized nanobody. To expand nanobody-based targeting to this or any other organelle, a few epitope tags with matching nanobodies can be employed, including the synthetic 13 amino acid ALFA tag (*Götzke et al., 2019*), the C-tag derived from α-synuclein (*De Genst et al., 2010*), and the SPOT-tag derived from β-catenin (*Traenkle et al., 2015*). Targeting via these tags is not only orthogonal to EGFP or mCherry-based targeting, but also expected to only exhibit a minimal effect on the tagged protein. This may be particularly advantageous if nanobody binding to a fluorescent reporter protein or direct fusion to a fluorescent protein alters its optical properties. This has to be carefully evaluated for each protein of interest. Importantly, epitope tags are also compatible with the reverse strategy: endogenous ciliary proteins can be tagged by CRISPR/Cas9-based genome-editing techniques, allowing optogenetic tools or sensors, fused to the respective nanobodies, to be recruited to ciliary proteins expressed at endogenous levels, and perhaps even to specific subdomains of primary

cilia. Along the same lines, nanobodies against endogenous cilia proteins, for example Arl13b may reduce the necessary genetic modifications for cilia targeting. Nanobody-based binding controlled by light or by small molecules will further expand the versatility of this approach (*Yu et al., 2019*; *Gil et al., 2019*; *Farrants et al., 2020*). To date, all intracellular applications of nanobodies require genetic modification of target cells, a limitation that may be overcome with improved strategies for intracellular protein delivery.

Since several proteins proposed to function in cilia also have well-known functions outside the cilium (such as PKA or AMPK), it is a major challenge in cilia biology to specifically interfere with their cilium-specific functions. One approach is to target proteins or peptides with inhibitory functions specifically to the cilium, for example to interfere with PKA activity inside cilia (*Mick et al., 2015*). Cilia length is an important parameter that determines cilia function (*Hsu et al., 2017*). However, the spatial contribution of signaling pathways in the cilium or cell body that regulate cilia length is not well understood. So far, only pharmacology that lacks spatial resolution has been used to investigate the role of cAMP in the regulation of cilia length. Using our approach to optogenetically manipulate cAMP levels in the cilium or cell body allows to address the spatial contribution of cAMP signaling in cilia length control. Our results indicate that an increase in cAMP levels and, thereby, cAMP signaling in the cilium or the cell body exerts opposing effects by either increasing or decreasing cilia length, respectively (*Figure 6H*). Indeed, it has been demonstrated in HEK293 cells that an increase in cellular cAMP levels and concomitant PKA-dependent protein phosphorylation at centriolar satellites induced ubiquitination and proteolysis of NEK10 by the co-assembled E3 ligase CHIP and promoted cilia resorption (*Porpora et al., 2018*). This is supported by elegant studies of cilia length control in *Chlamydomonas* (*Wang et al., 2019*; *Hendel et al., 2018*; *Ishikawa and Marshall, 2017*; *Liang et al., 2018*; *Ludington et al., 2015*; *Cao et al., 2013*; *Hu et al., 2015*; *Luo et al., 2011*; *Meng and Pan, 2016*; *Pan and Snell, 2005*). Here, the IFT machinery and its regulation by phosphorylation plays an important role. There is a negative correlation between IFT particles entering flagella and flagella length, suggesting a length-dependent feedback control of IFT entry. At least in *Chlamydomonas*, phosphorylation of the kinesin-II subunit Fla8 (pFla8) by the calcium-dependent kinase 1 (CDPK1), a homolog of CAMKII, changes the rate of IFT entry and, thereby, flagella length (*Liang et al., 2018*). Thus, a cellular sensing system controls pFla8 levels, reduces the rate of IFT entry and controls flagella length. In contrast, stimulation of cAMP signaling in the cilium results in an increase in cilia length, probably through cAMP/PKA-dependent signaling pathways, as suggested by experiments in primary cilia of epithelia and mesenchymal cells, where cilia elongation is induced by a cAMP/PKA-dependent mechanism (*Besschetnova et al., 2010*). In line with this finding, an increase in intracellular cAMP levels and downstream activation of PKA has been shown to contribute to an increase in cilia length by regulating anterograde IFT transport velocity (*Besschetnova et al., 2010*). In these reports, cells have been pharmacologically stimulated with for example Forskolin to increase intracellular cAMP levels. Forskolin activates transmembrane ACs, which have been identified in both, the cilium and cell body. Thus, Forskolin evokes an increase of the cAMP concentration in the cilium and the cell body. Based on the data presented here, this could outweigh the respective effect of cAMP in the cilium or cell body on ciliary length control, or, depending on the expression level of ACs in either compartment, favor an increase or decrease of the ciliary length, respectively. This may explain why some reports demonstrate an increase (*Besschetnova et al., 2010*), while others see a decrease in cilia length upon Forskolin stimulation (*Porpora et al., 2018*). We also performed stimulation with Forskolin and detected no significant change in ciliary length because the data was rather variable (*Figure 6A*). Thus, experiments using pharmacological stimulation of cAMP levels have to be interpreted with caution.

Although a light-stimulated increase in ciliary cAMP levels by bPAC resulted in an increase in cilia length, a decrease in ciliary cAMP levels evoked by photoactivation of LAPD did not change cilia length. Only a cAMP increase in the cell body resulted in a reduction of the cilia length. Thus, cilia length control seems to rely on a spatially segregated increase, but not decrease of cAMP levels. The origin of the cAMP increase determines whether cilia become shorter or longer. A possible explanation for this surprising finding is that PKA is the major target for cAMP and has also been implicated in the cAMP-dependent regulation of cilia length control. In vitro, PKA is half-maximally activated at 100–300 nM. However, a recent report demonstrated that in fact, the sensitivity of PKA for cAMP is almost twenty times lower in cells compared to in vitro (*Koschinski and Zaccolo, 2017*). In many different cell types, resting cAMP levels have been shown to be around 1 µM

(*Koschinski and Zaccolo, 2017*), and primary cilia in mIMCD-3 cells contain a similar cAMP concentration as the cell body (*Jiang et al., 2019*). Based on these numbers, we hypothesize that cilia length is mainly regulated by PKA-dependent mechanisms, only responding to an increase of the local cAMP concentration. As described above, this increase has to be well above >1 μM to activate PKA in cells. In turn, reducing basal cAMP levels below 1 μM by photoactivation of LAPD would not result in a change in ciliary length since the activity of PKA, compared to basal levels, would not be changed.

Our method is highly versatile, as it allows to both manipulate and measure cAMP signaling. Thus, ciliary signaling cascades involving cAMP would be a prime subject to investigate in future experiments. In fact, this includes the prominent ciliary Hedgehog signaling pathway, as stimulation with Sonic Hedgehog (Shh) has been proposed to reduce ciliary cAMP levels (*Moore et al., 2016*). Furthermore, GPCR signaling pathways use cAMP as a second messenger for signal transduction. Primary cilia are enriched in specific GPCRs, for example 5-HT$_6$, V2R, EP4, SSTR3, or GPR120 (*Jin et al., 2014*; *Berbari et al., 2008*; *Nachury and Mick, 2019*; *Sherpa et al., 2019*; *Hilgendorf et al., 2019*; *Hilgendorf et al., 2016*). Thus, the function of these GPCR-dependent signaling pathways could be analyzed with spatial and temporal resolution using our approach.

Our approach and complementary developments in other labs using nanobodies for subcellular targeting (*Herce et al., 2013*; *Harmansa et al., 2017*; *Beghein et al., 2016*; *Steels et al., 2018*) open up new avenues for analyzing signaling pathways in primary and motile cilia, as demonstrated by the application of the cilia-targeted nanobody in zebrafish, and beyond that in many other subcellular domains in vitro and in vivo. The nanobody-based targeting approach conveys high specificity through a strong interaction with its binding partner ($K_d$ ~1 nM), which is a prerequisite for subcellular targeting and binding of endogenously expressed proteins in vitro and in vivo (*Beghein and Gettemans, 2017*; *Van Overbeke et al., 2015*). In addition, nanobodies are small (~15 kDa) and genetically encoded on one single open reading frame (*Beghein and Gettemans, 2017*). Thus, the generation of transgenic animals, for example mice or zebrafish, expressing the nanobody targeted to a subcellular compartment is straightforward. Analogous to the Cre/loxP recombinase system, the nanobody-based targeting approach offers endless combinations of targeting any protein of interest to a desired subcellular compartment by simply crossing different transgenic lines. This will greatly facilitate the analysis of cellular signaling in the whole cell and in a specific compartment in vivo. Conditional expression patterns will further allow temporally controlled recruitment or cell-type-specific localization. Many transgenic animals have already been generated, expressing optogenetic tools that are either fused to a fluorescent protein or even contain fluorescent proteins as functional read-out, for example biosensors for second messengers. Combining these transgenic lines with transgenic animals that allow nanobody-based subcellular targeting will pave the way for a more systematic and nuanced application of optogenetics to study cellular signaling, for example by targeting Ca$^{2+}$, PKA activity, or lipid sensors, enzymes, or even transcription factors to the primary cilium. Such studies will unravel how subcellular compartments control cellular functions under physiological and pathological conditions.

# Materials and methods

**Key resources table**

| Reagent type (species) or resource | Designation | Source or reference | Identifiers | Additional information |
|---|---|---|---|---|
| Gene (mouse) | Nphp3 | NCBI | NM_028721 | |
| Cell line (human) | HEK293 | ATCC | CRL-1573 | |
| Cell line (human) | HEK-TM | *Wachten et al., 2006* | | |
| Cell line (mouse) | mIMCD-3 | ATCC | CRL-2123 | |
| Transfected construct (mouse) | See Sup. Table 1 | See Sup. Table 1 | See Sup. Table 1 | |
| Biological sample (*Danio rerio*) | nacre | *Lister et al., 1999* | | |

*Continued on next page*

*Continued*

| Reagent type (species) or resource | Designation | Source or reference | Identifiers | Additional information |
|---|---|---|---|---|
| Biological sample (*Danio rerio*) | *b-actin:arl13b-gfp* | *Borovina et al., 2010* | | |
| Biological sample (*Danio rerio*) | *Ubi:zebrabow* | *Pan et al., 2013* | | |
| Antibody (mouse) | anti-acetylated-Tubulin (mouse, monoclonal) | Sigma Aldrich | T6793 | 1:600 |
| Antibody (rabbit) | anti-GFP (rabbit, polyclonal) | Abcam | ab6556 | 1:500 |
| Antibody (rabbit) | anti-Arl13B (rabbit, polyclonal) | Proteintech | 17711–1-AP | 1:500 |
| Antibody (mouse) | anti-Arl13B (mouse, monoclonal) | Abcam | ab136648 | 1:500 |
| Antibody (donkey) | anti-mouse-Cy5 (donkey) | Dianova | 715-175-151 | 1:500 |
| Antibody (goat) | anti-rabbit-Alexa488 (goat) | Life Technologies | A11034 | 1:500 |
| Recombinant DNA reagent | cADDis cAMP assay kit | Montana Molecular | #D0201/11G | |
| Sequence-based reagent (IDT) | VHH$_{LaM-2,}$ VHH$_{LaM-4}$ | Gene blocks (IDT) *Fridy et al., 2014* | | |
| Sequence-based reagent (Ploegh lab) | VHH$_{Enhancer}$ | Hidde Ploegh, Boston, USA *Kirchhofer et al., 2010* | | |
| Commercial assay or kit (Thermo) | Lipofectamine 2000 | Thermo Fisher Scientific | #11668030 | |
| Commercial assay or kit (Molecular Devices) | CatchPoint assay | Molecular Devices | | |
| Commercial assay or kit (Thermo) | Pierce BCA Protein Assay Kit | Thermo Fisher Scientific | 23227 | |
| Commercial assay or kit (Qiagen) | QIAquick PCR purification kit | Qiagen | #28104 | |
| Commercial assay or kit (Thermo) | mMessage mMachine T7 kit | Thermo Fisher Scientific | #AM1344 | |
| Chemical compound, drug (Sigma) | PEI | Sigma Aldrich | #64604–1G | |
| Chemical compound, drug (Merck) | ChemiBLOCKER | Merck Millipore | #2170 | |
| Chemical compound, drug (Polysciences) | Aqua-Poly/Mount | Polysciences | #18606–20 | |
| Chemical compound, drug (Thermo) | DAPI | Thermo Fisher Scientific | D1306 | 1:10.000 |
| Chemical compound, drug (Enzo) | FluoForte | Enzo Life Sciences | ENZ-52015 | |
| Chemical compound, drug (Sigma) | Fluo4-AM | Sigma Aldrich | 93596 | |
| Chemical compound, drug (Thermo) | probenecid | Thermo Fisher Scientific | P36400 | |
| Chemical compound, drug (Sigma) | NKH477 | Sigma Aldrich | N3290 | |
| Chemical compound, drug (Sigma) | Forskolin | Sigma Aldrich | F3917 | |
| Chemical compound, drug (Sigma) | IBMX | Sigma Aldrich | I5879 | |

*Continued*

| Reagent type (species) or resource | Designation | Source or reference | Identifiers | Additional information |
|---|---|---|---|---|
| Software, algorithm (this paper) | CiliaQ | This publication | https://github.com/hansenjn/CiliaQ; *Hansen, 2020*; copy archived at https://github.com/elifesciences-publications/CiliaQ | |

## Plasmids

Coding sequences for codon-optimized anti-mCherry nanobodies VHH$_{LaM-2}$ and VHH$_{LaM-4}$ were synthesized as GeneBlocks by IDT, based on the amino acid sequences provided by *Fridy et al., 2014*. A vector encoding the anti-eGFP nanobody GBP-1 (VHH$_{Enhancer}$) (*Kirchhofer et al., 2010*) was kindly provided by the lab of Hidde Ploegh (Boston Children's Hospital, Boston, MA). The cDNA sequence encoding the anti-mCherry nanobodies was fused to the 3' end of the sequence encoding amino acid 1–201 of mNphp3 (NCBI NM_028721, release 16/09/2018), and to eGFP or an hemagglutinin HA-tag at the 3' end. Similarly, the cDNA encoding the anti-eGFP nanobody was fused to the 5' end of the sequence encoding amino acid 1–201 of mNphp3 and at the 3' end to an HA-tag. The respective constructs were cloned into the pcDNA3.1(+) vector (Thermo Fisher Scientific) for expression in mammalian cells. All primer sequences used for cloning and the corresponding plasmids are summarized in Supplementary Table 1.

## Cell lines and tissue culture

HEK293 (CRL-1573) and mIMCD-3 (CRL-2123) cells were obtained and authenticated from American Type Culture Collection (ATCC). HEK293 TM (HEK-TM) cells were generated as described previously (*Wachten et al., 2006*). HEK-TM cells were transfected with pc3.1-bPAC-mCherry or pcDNA6-LAPD-mCherry and selected for stable expression. HEK-mlCNBD-FRET were generated as described previously (*Mukherjee et al., 2016*). HEK293 cells were maintained in Dulbecco's modified Eagle's medium (DMEM) (Gibco), supplemented with 1x GlutaMax (Gibco) and 10% Fetal Calf Serum (FCS) (Biochrome) at 37°C and 5% $CO_2$ atmosphere. mIMCD-3 cells were maintained in DMEM/F12 (1:1) medium, supplemented with GlutaMax and 10% FCS at 37°C and 5% $CO_2$. Additionally, individual media contained the following: HEK-TM cells: 0.1 mg/ml hygromycin (Thermo Fisher Scientific), HEK-TM bPAC-mCherry: 0.1 mg/ml hygromycin (Thermo Fisher Scientific) and 0.8 mg/ml G418 (Thermo Fisher Scientific), HEK-TM LAPD-mCherry: 50 µg/ml hygromycin, 5 µg/ml blasticidin (Thermo Fisher Scientific), HEK-TM mNphp3(201)-LAPD-mCherry: 0.1 mg/ml hygromycin (Thermo Fisher Scientific) and 0.8 mg/ml G418 (Thermo Fisher Scientific), HEK-mlCNBD-FRET cells: 0.8 mg/ml G418 (Thermo Fisher Scientific), mIMCD-3 bPAC-mCherry: 0.8 mg/ml G418 (Thermo Fisher Scientific), mIMCD-3 LAPD-mCherry: 5 µg/ml blasticidin (Thermo Fisher Scientific). During the experiments, cells were kept without antibiotics. All cells have been tested and free from mycoplasma and other microorganisms.

## Transfection

mIMCD-3 cells were transfected with Lipofectamine 2000 (Thermo Fisher Scientific) and HEK293 cells with polyethylenimine (PEI, Sigma Aldrich). For transfection with Lipofectamine 2000 Reagent, the transfection medium was replaced after 4–5 hr with full medium. For PEI transfection (four-well dish), 0.5 µg plasmid DNA per well was mixed with 1 µg PEI in 50 µl OptiMEM (Gibco), incubated at room temperature for 10 min, and added to 200 µl full medium on the cells. For PEI transfection (96-well plate), 0.1 µg DNA was mixed with 0.2 µg PEI in 10 µl OptiMEM, incubated at room temperature for 10 min, and added to the cells in 50 µl full medium containing 2% FCS. All cells referred to as non-transfected (NT) were subjected to the same transfection protocol as transfected cells, but without adding DNA, Lipofectamine and PEI.

## Immunocytochemistry

Immunocytochemistry was performed according to standard protocols. Cells were seeded on poly-L-lysine (PLL, 0.1 mg/ml, Sigma Aldrich)-coated 13 mm glass coverslips (VWR) in a four-well dish (VWR) and transfected on the next day as described above. For mIMCD-3 cells, the medium was

replaced with starvation medium (0.5% FCS) on the next day to induce ciliogenesis. Cells were fixed 24 hr after inducing ciliogenesis (mIMCD-3) or 24–48 hr after transfection (HEK293) with 4% paraformaldehyde (Alfa Aesar, Thermo Fisher Scientific) for 10 min at room temperature. After washing with PBS, cells were blocked with CT (0.5% Triton X-100 (Sigma Aldrich) and 5% ChemiBLOCKER (Merck Millipore) in 0.1 M NaP, pH 7.0) for 30 min at room temperature. Primary and secondary antibodies were diluted in CT and incubated for 45 and 60 min at room temperature, respectively. Coverslips were mounted with one drop of Aqua-Poly/Mount (Tebu-Bio).The following antibodies were used: mouse anti-acetylated-Tubulin (1:600, Sigma Aldrich, T6793), rabbit anti-GFP (1:500, Abcam, ab6556, IgG), mouse anti-Arl13B (Abcam, ab136648, 1:500), rabbit anti-Arl13B (1:500, Proteintech, 17711–1-AP), donkey anti-mouse-Cy5 (1:500, Dianova, 715-175-151), goat anti-rabbit-Alexa488 (1:500, Life Technologies, A11034). As a DNA counterstain, DAPI was used (4′,6-Diamidino-2-Phenyl-indole, Dihydrochloride, 1:10 000, Invitrogen) and cells were mounted in Aqua-Poly/Mount (Tebu-Bio).

## Optogenetic stimulation for cilia length measurements

mIMCD-3 cells and mIMCD-3 bPAC cells were seeded, transfected with pcDNA3.1-mNphp3(201)-VHH$_{LaM-2}$-eGFP, and induced to form cilia as described above. Cells were kept in the dark during the entire experiment and handled only under dim red (bPAC) or green (LAPD) light, preventing bPAC- or LAPD-activation, respectively. For 'light' stimulation, cells were placed on a LED plate (bPAC: 465 nm, 38.8 µW/cm$^2$; LAPD: 630 nm, 42.3 µW/cm$^2$) for the last 16 hr before harvesting or for 1 hr at 24 hr prior to harvesting (as indicated in the figure legends). Cells were fixed and further analyzed by immunocytochemistry and confocal microscopy.

## Confocal microscopy and image analysis

Confocal z-stacks (step size 0.4–0.5 µm, 60x objective) were recorded with a confocal microscope (Eclipse Ti, Nikon or Olympus FV100). All depicted images show a maximum projection of a z-stack unless differently stated in the figure legend. For quantifying cilia length and fluorescence signals, z-stacks were recorded from at least two (for stable cell lines) or three (for transiently transfected cells) random positions per experiment and analyzed using custom-written ImageJ plug-ins. Channels were split and the channel representing the Arl13B-staining was segmented by applying an intensity threshold calculated in a maximum projection of the channel ('RenyiEntropy'-algorithm, implemented in ImageJ). The segmented Arl13B channel served as a mask for cilia in the other channels, which were then subjected to a custom-written ImageJ plug-in called 'CiliaQ'. We developed CiliaQ to fully-automatically quantify the cilia length and ciliary intensity levels in the different channels. CiliaQ detects individual 3D objects in the segmented channel and filters out 3D objects below a pre-defined size threshold (10 voxel) to exclude noise. Each remaining 3D object is considered as a cilium. For each cilium, CiliaQ determines the average intensity of all pixels belonging to the 3D region in each channel. The length of the cilium is determined as the length of the ciliary 3D skeleton obtained by skeletonizing (*Arganda-Carreras et al., 2010*) the three-fold-upscaled and blurred (Gaussian blur, sigma = 3 corresponding to 0.21 µm) image of the corresponding ciliary 3D region. All results were scrutinized by a trained observer.

## Ca$^{2+}$ imaging

Ca$^{2+}$ imaging in 96-well plates in a fluorescence plate reader was performed as previously described (*Jansen et al., 2015*; *Stabel et al., 2019*). For probing LAPD activity, HEK-TM or HEK-TM-LAPD cells were seeded on a PLL (0.1 mg/ml, Sigma Aldrich)-coated 96-well plate (F-Bottom, CELLSTAR, Greiner) at $3 \times 10^4$ cells per well and incubated over night at 37°C and 5% CO$_2$ in darkness. For probing bPAC activity, HEK-TM or HEK-TM-bPAC cells were seeded on a PLL (0.1 mg/ml, Sigma Aldrich)-coated 96-well plate (F-Bottom, CELLSTAR, Greiner) at $4 \times 10^4$ cells per well, HEK-TM cells were PEI-transfected with pc3-mNphp3(201)-bPAC-mCherry or pcDNA3.1zeo_mCherry on the next day, and incubated over night at 37°C and 5% CO$_2$ in darkness. For probing LAPD or bPAC activity during nanobody binding, HEK-TM, HEK-TM-LAPD, or HEK-TM-bPAC cells were as described for LAPD activity measurements, transfected on the next day with pEGFP-N1_sstr3, pcDNA3.1-mNphp3(201)-VHH$_{LaM-2}$-eGFP, pcDNA3.1-mNphp3(201)-VHH$_{LaM-2}$-HA, pcDNA3.1-mNphp3(201)-VHH$_{LaM-4}$-eGFP, or pcDNA3.1-mNphp3(201)-VHH$_{LaM-4}$-HA using PEI transfection, and incubated over night at

37°C and 5% $CO_2$ in darkness. All following steps were conducted under dim green light (LAPD) or dim red light (bPAC). Medium was removed, and cells were washed with 50 µl ES (extracellular solution) buffer (120 mM NaCl, 5 mM KCl, 2 mM $CaCl_2$, 2 mM $MgCl_2$, 10 mM glucose, 10 mM HEPES pH 7.4). Cells were loaded with 2 µM FluoForte (bPAC, bPAC+nanobody, LAPD+nanobody, Enzo Life Sciences) or 2 µM Fluo4-AM (LAPD) (stocks in DMSO/Pluronic F-127 (Sigma-Aldrich)) and 3 mM probenecid (Invitrogen) in 50 µl ES for 30 min at 37°C. Afterwards, the buffer was replaced with 90 µl ES containing 3 mM probenecid, and cells were incubated for 30 min at 29°C in a fluorescence plate-reader (FLUOstar omega, BMG Labtech). Fluorescence was measured at 29°C with an Ex544 excitation and a 570 ± 10 nm emission filter (FluoForte) or with a 485 ± 6 nm excitation and a 530 ± 15 nm emission filter (Fluo4) (all filters BMG Labtech). During bPAC activity measurements, cells were stimulated with a 488-nm-light pulse (1 W/cm$^2$) for 1 s at 3 min, for 10 s at 21 min, and for 60 s at 45 min. For bPAC+nanobody activity measurements, cells were incubated with 25 µM of IBMX (stock: 250 mM in DMSO, AppliChem) from 5 min before light activation on; light activation was induced 2 min after starting the recording using a 488-nm-light pulse (1 s, 162 µW/cm$^2$). For LAPD and LAPD+nanobody activity measurements, cells were stimulated after 2 min with 100 µM NKH477 (Sigma-Aldrich) in ES buffer. During the measurement, the plate was illuminated with an 850 nm LED (0.5 µW/cm$^2$) inside the reader and then switched to a 690 nm LED (0.5 µW/cm$^2$) to activate LAPD. At the end of all experiments, ionomycin was added (final concentration: 2 µM, stock: 1 mM in DMSO, Tocris), and fluorescence was recorded until saturation of the signal amplitude. After the end of recording, cell integrity and transfection rate was scrutinized by confocal microscopy of the recorded wells.

## FRET imaging

FRET imaging was performed as previously described (*Mukherjee et al., 2016*). HEK293 or HEK-mlCNBD-FRET cells were seeded and transfected with pcA-Cerulean, pcA-Citrine, pcA-Cerulean + pc3.1-VHH$_{enhancer}$-mCherry, or pcA-Citrine + pc3.1-VHH$_{enhancer}$-mCherry (HEK293), and pc3.1-VHH$_{enhancer}$-mCherry, pcDNA3.1zeo_mCherry (HEK-mlCNBD-FRET) as described for LAPD activity measurements. Fluorescence imaging of live cells was performed using the CellR Imaging System (Olympus), consisting of an inverse, fully motorized wide-field microscope (IX81) with a monochromatic CCD camera (XM10), a reflector turret, and an illumination system with an excitation-filter wheel (MT20, 150 W Xenon arc burner). Measurements were performed with a 20x/0.75 objective (UPlanSApo, Olympus) at room temperature under atmospheric conditions. Before the measurement, cells were washed once with ES (extracellular solution) buffer and measurements were performed in ES buffer. The experimental recordings were as follows: Before and after each time-resolved measurements, the mCherry fluorescence (12% light intensity, 200 ms exposure time, 575/25 excitation filter, mCherry-B-0MF Semrock dichroic mirror, 630/20 emission filter) was measured. Time-resolved measurements captured the cerulean (12% light intensity, 100 ms exposure time, 430/25 excitation filter, M2CFPYFP Olympus dichroic mirror, 480/40 emission filter), the citrine fluorescence (12% light intensity, 100 ms exposure time, 500/20 excitation filter, M2CFPYFP dichroic mirror, 535–30 emission filter), and the FRET signal (12% light intensity, 100 ms exposure time, 430/25 excitation filter, M2CFPYFP dichroic mirror, 535–30 emission filter) every 5 s. After 120 s, cells were stimulated with 20 µM isoprenaline hydrochloride (isoproterenol, Sigma Aldrich) or ES buffer as a control. Data was analyzed using Fiji/ImageJ (ImageJ Version 1.52i) (*Schindelin et al., 2012*) by selecting mCherry positive cells with freehand ROIs and determining the mean fluorescence intensity for each ROI in each channel. Values were background subtracted and the FRET signal was corrected for bleed-through and cross-excitation with the following formula: $FRET_{corrected} = FRET - α * cerulean - β * citrine$ (with α being the determined bleed-through constant and β being the determined cross-excitation constant for the chosen experimental set-up; α and β values were calculated from single cerulean or citrine transfected cells of three independent experiments using the innate ImageJ tool 'Coloc 2', and are 0.75 and 0.02, respectively). Data were plotted as a change of cerulean/FRET$_{corrected}$ over time. Data were acquired from n = 3 independent experiments.

## R-FlincA imaging

HEK293 cells were seeded and transfected with pcDNA4HMB_R-FlincA or pcDNA4HMB_R-FlincA-mut (R221E, R335E) (*Ohta et al., 2018*) (generously provided by Kazuki Horikawa, Tokushima

University, Japan) and pEGFP-N1-bPAC (see Supp. Table 1) as described for LAPD activity measurements. Imaging was performed using the CellR Imaging System (Olympus). The experimental recordings were as follows: R-FlincA signal (57% light intensity, 200 ms exposure time, 572/25 excitation filter, mCherry-B-0MF Semrock dichroic mirror, 630/20 emission filter) was measured every 5 s. At 120 s, cells were illuminated for 5 s with 2.1 mW/cm$^2$ white light, followed by further recording of R-FlincA signal every 5 s for 480 s. Subsequently, bPAC-GFP fluorescence (12% light intensity, 100 ms exposure time, 500/20 excitation filter, M2CFPYFP dichroic mirror, 535–30 emission filter) was measured. Data were analyzed using Fiji/ImageJ (ImageJ Version 1.52i) by selecting low-expressing bPAC-GFP cells with freehand ROIs and determining the mean fluorescence intensity for each ROI in the average signal recorded during the 120 s before white light exposure. Data were plotted as a change of fluorescence over time. Data were acquired from n = 3 measurements.

## Imaging of primary cilia

mIMCD-3 cells were seeded on PLL (0.1 mg/ml, Sigma Aldrich)-coated chambers (µ-Slide 8 Well Glass Bottom, ibidi) and transfected after 24 hr with pc3.1-VHH$_{enhancer}$-HA and pc3.1-mlCNBD-FRET (*Mukherjee et al., 2016*) as described above. The medium was replaced with starvation medium (0.5% FCS) on the following day to induce ciliogenesis. Confocal FRET imaging was performed at the DZNE Light Microscopy Facility using the Andor Spinning Disk Setup (built on an inverted Eclipse Ti Microscope, Nikon) at 37° C. For FRET imaging, the 445 nm laser (18% intensity, 445-, 514-, 640-triple dichroic mirror in the Yokogawa CSU-X1 unit and 5000 rpm disk speed) was used as excitation source, combined with a dual-cam CFP/YFP filter cube (509 nm dichroic mirror with 475/25 nm and 550/49 nm emission filters) to simultaneously measure cerulean and citrine emission with the two EM-CCD cameras (100 ms exposure time, 300 EM gain, 5.36 frames per second frame rate, 10.0 MHz horizontal readout, 1.7 µs vertical readout time, 5x pre Amp gain, −70° C camera temperature). The imaging procedure was as follows: Cells were washed once with ES buffer and measurements were performed in ES buffer. Cilia were imaged with a 100x/1.45 oil objective with 1 µm step size in 10 s intervals. After a stable baseline was obtained, cells were stimulated by drug addition. Cilia-specific fluorescence values were obtained by analyzing the recordings using CiliaQ as described above in 'Confocal microscopy and image analysis'. The FRET signal was calculated as a ratio of cerulean/citrine, normalized to the mean baseline value before stimulus addition, and plotted as a change over time.

## cADDis imaging

mIMCD-3 cells were seeded as described above. After 24 hr, cells were transduced with the ratiometric cilia-targeted cADDis cAMP assay kit (5-HT$_6$-mCherry-cADDis, Montana Molecular). In detail, 25 µl of the BacMAM stock was mixed with 3 µl sodium butyrate (Sigma Aldrich), and 22 µl Opti-MEM (ThermoFischer Scientific). The growth medium on the cells was exchanged with 250 µl Opti-MEM, and the 50 µl mixture containing BacMAM was added dropwise to the well. Cells were incubated for 24 hr at 37° C, 5% CO$_2$ and subsequently measured at the DZNE Light Microscopy Facility using the Andor Spinning Disk Setup. For ratiometric cADDis imaging, cpGFP and mCherry were excited with the 448 nm (10%) and 561 nm (10%) lasers, respectively, in combination with a 405, 448-, 561-, 640-quad dichroic mirror in the Yokogawa CSU-X1 unit and 5000 rpm disk speed. Images were acquired on the two EM-CCD cameras simultaneously (100 ms 100 ms exposure time, 300 EM gain, 5.36 frames per second frame rate, 10.0 MHz horizontal readout, 1.7 µs vertical readout time, 5x pre Amp gain, −70° C camera temperature) with a GFP/RFP emission filter cube (580 nm LP dichroic and 617/73 nm and 525/50 nm emission filters). The experimental procedure during imaging and data analysis was performed as described in the Materials and methods section ' Imaging of primary cilia'. Data analysis was as described above form mlCNBD-FRET sensor imaging, but without correction for bleed-through and cross-excitation. Accordingly, the cADDis signal was plotted as a ratio of mCherry/cpGFP, normalized to the mean baseline value before stimulus addition, and plotted as a change over time.

For imaging in combination with bPAC, mIMCD3 cells were transfected with mNphp3(201)-VHH$_{Lam2}$-HA and bPAC-mCherry or mCherry as described above. After 24 hr, cells were transduced with the cilia-targeted cADDis cAMP assay kit (5-HT$_6$-cADDis, Montana Molecular) as described above. Six hours after transduction, the medium was replaced with 250 µl starvation medium per

well containing 2 µM sodium butyrate and further incubated at 37° C overnight. 24 hr post transduction, cells were measured at the Microscopy Core Facility of the Medical Faculty at the University of Bonn using the Visitron VisiScope Spinning Disk Setup (Build on a Zeiss Axio Observer, Zeiss) at 37° C. For non-ratiometric cADDis imaging in combination with bPAC-mCherry, cpGFP and mCherry were excited with the 448 nm (8%) and 561 nm (10%) laser, respectively, in combination with the 405-488-560bs dichroic mirror in the Yokogawa CSU-W1 unit and the 50 µm pinhole disk at 4000 rpm disk speed. A 60x C-Apochromat water objective (NA = 1.2) was used. Images were acquired on two pco.edge sCMOS cameras simultaneously (200 ms exposure time, 2x binning) with a GFP/RFP emission filter cube. The experimental procedure during imaging and image analysis was the same as described above.

## ELISA-based cAMP measurements

Total cAMP levels were determined using a CatchPoint assay (Molecular Devices) according to manufacturer's instructions mIMCD-3, mIMCD-3 bPAC-mCherry, or mIMCD-3 LAPD-mCherry cells were seeded on a PLL (0.1 mg/ml, Sigma Aldrich)-coated 96-well plate (F-Bottom, CELLSTAR, Greiner) at $1.8 \times 10^4$ cells per well and incubated over night at 37°C and 5% $CO_2$ in darkness. During all further experimental procedures, cells or cell lysates were kept in the dark and handled only under dim red (bPAC) or green (LAPD) light, preventing bPAC- or LAPD-activation, respectively. After 48 hr, the medium was changed to starvation medium (0.5% FCS) to induce ciliogenesis. Another 24 hr later, the medium was replaced with ES and cells were either subjected to a 2 min light pulse (465 nm, 38.8 µW/cm$^2$; LAPD: 630 nm, 42.3 µW/cm$^2$) or kept in the dark. Directly after light stimulation, cells were lysed and cAMP amounts per well were determined using a CatchPoint assay (Molecular Devices), according to manufacturer's instructions. The protein concentration per well was determined with a Pierce BCA Protein Assay Kit (Thermo Fisher Scientific), according to manufacturer's instructions.

## Zebrafish as an experimental model

The animal facilities and maintenance of the zebrafish, *Danio rerio*, were approved by the Norwegian Food Safety Authority (NFSA, 19/175222). Fishes were kept in 3.5 l tanks in a Techniplast Zebtech Multilinking system at 28°C, pH 7 and 700 mSiemens, at a 14/10 h light/dark cycle. Fish were fed dry food (ZEBRAFEED; SPAROS I and D Nutrition in Aquaculture) two times/day and *Artemia nauplii* once a day (Grade0, platinum Label, Argent Laboratories, Redmond, USA). Embryos were maintained in egg water (1.2 g marine salt and 0.1% methylene blue in 20 l RO water) from fertilization to imaging. All procedures were performed on zebrafish embryos in accordance with the directive 2010/63/EU of the European Parliament and the Council of the European Union and the Norwegian Food Safety Authorities. For experiments the following zebrafish lines were used: nacre (*mitfa-/-*) (**Lister et al., 1999**), b-actin:arl13b-gfp (**Borovina et al., 2010**), and *Ubi:zebrabow* (**Pan et al., 2013**) transgenic animals, which express RFP ubiquitously in absence of Cre recombinase.

## mRNA synthesis, injection, immunostaining, and imaging

The mRNA synthesis, injection, immunostaining, and imaging was performed according to standard protocols. In order to generate capped mRNA of the cherry-nanobody, 5 µg of the plasmid pc3.1-mNphp3 (201)-VHH$_{LaM-2}$-eGFP was first linearized using FastDigest BpiI for 30 min at 37°C (Thermo Fisher Scientific, Cat# FD1014). Upon verification of the linearization of the plasmid by gel electrophoresis, the digested plasmid was purified using the QIAquick PCR purification kit (Qiagen, cat #28104) and its DNA concentration measured using a Nanodrop spectrophotometer (Thermo Fisher Scientific). mRNA was in vitro transcribed from 500 ng of linearized plasmid using the mMessage mMachine T7 kit according to the supplier's instructions (Thermo Fisher Scientific, Cat # AM1344). Following 3 hr incubation at 37°C, the mRNA was precipitated upon addition of 40 µl RNase-free $H_2O$ and 30 µl LiCl precipitation solution provided in the mMessage mMachine T7 kit, incubation overnight at −20°C, and centrifugation for 30 min at 13.000 rpm at 4°C. The precipitated mRNA was further washed with 70% Ethanol in RNase-free $H_2O$, air dried, and resuspended in RNase-free $H_2O$. The integrity of the mRNA was confirmed by gel electrophoresis and its RNA concentration measured with a nanodrop spectrophotometer. The mRNA was aliquoted and kept at −80°C until injection.

200–300 pg of capped cherry-nanobody mRNA diluted in 0.2M KCl and 0.5% phenol red was injected in the yolk of one-cell stage embryos using a pressure microinjector (Eppendorf Femtojet 4i). Injection needles were pulled with a Sutter Instrument Co. Model P-2000, from thin-walled glass capillaries (1.00 mm; VWR), using the following settings: heat = 450, filament = 4, velocity = 50, delay = 225, pull = 100. The needle tip was cut open with forceps. The pressure and time used for the injection were calibrated for each needle using a 0.01 mm calibration slide for microscopy and a drop of mineral oil to perform injections of 1–2 nl.

Dechorionated and euthanized embryos (collected between 24 and 26 hpf) were fixed in 4% paraformaldehyde solution and 1% DMSO for 2 hr at room temperature. Embryos were washed three times 5 min with 0.3% Triton-X-100 in PBS (PBSTx), permeabilized with 100% acetone for 10 min at −20°C, washed three times 10 min with 0.3% PBSTx and blocked in 0.1% BSA/0.3% PBSTx for 2 hr. Embryos were incubated with an anti-acetylated tubulin antibody (6-11B-1, 1:1000, Sigma Aldrich) overnight at 4°C, subsequently washed three times 1 hr with 0.3% PBSTx, and incubated with the secondary antibody (Alexa-labelled GAM555 plus, Thermo Fisher Scientific, 1:1000) and an Alexa Fluor 488-coupled anti-GFP antibody (A-21311, 1:1000, Thermo Fisher Scientific) overnight at 4°C. Next, the samples were incubated for 2 hr with 0.1% DAPI in 0.3% PBSTx (Life Technology), washed three times 1 hr with 0.3% PBSTx, and transferred to a series of increasing glycerol concentrations (25% and 50%). Stained larvae were stored in 50% glycerol at 4°C and imaged using a Zeiss Examiner Z1 confocal microscope with a 20x plan NA 0.8 objective.

Embryos at 22–26 hpf were manually dechorionated, anaesthetized in 0.01% pH 7.4 buffered MS-222, and mounted in 2% low-melting point agarose (dissolved in artificial fish water (1.2 g marine salt in 20 l RO water) in a Fluorodish (World Precision Instruments). Images were acquired using a Zeiss Examiner Z1 confocal microscope with a 20x water immersion NA 1.0 objective. Acquired images were processed with Fiji/ImageJ or Zen (Zeiss).

## Software

Data analysis and statistical analysis was performed in Excel (Microsoft Office Professional Plus 2013, Microsoft) and GraphPad Prism (Version 8.1.2, GraphPad Software, Inc). All image processing and analysis was performed in ImageJ (Version v1.52i, U.S. National Institutes of Health, Bethesda, Maryland, USA). Plots and Figures were generated using GraphPad Prism (Version 8.1.2, GraphPad Software, Inc) and Adobe Illustrator CS5 (Version v15.0.0, Adobe Systems, Inc). ImageJ plugins were developed in Java, with the aid of Eclipse Mars.2 (Release 4.5.2, IDE for Java Developers, Eclipse Foundation, Inc, Ottawa, Ontario, Canada).

## Code availability statement

The analysis workflow to study cilia length and fluorescence signal with its custom-written ImageJ plug-ins ('CiliaQ') is available through the following link https://github.com/hansenjn/CiliaQ (copy archived at https://github.com/elifesciences-publications/CiliaQ).

## Acknowledgements

The project was supported by grants from the Deutsche Forschungsgemeinschaft (DFG): SPP1926: grant MO2192/4-1 (to AM) and grant WA3382/2-1 (to DW), SPP1726: grant WA3382/3-1 (to DW), TRR83/SFB (to DW), FOR2743 (to DW), and under Germany's Excellence Strategy – EXC2151 – 390873048 (to DW and FIS), Emmy Noether (to FIS), SFB894/TP-A22 (to DUM), the Boehringer Ingelheim Fonds (to JNH), and a Samarbeidsorganet Helse Midt-Norge grant (to NJY). We thank Jens-Henning Krause for technical support, the Core Facility Nanobodies of the University of Bonn, the Microscopy Core Research Facility of the Bonn Technology Campus, the Core Research Facility for Light Microscopy (CRFS) of the DZNE (German Center for Degenerative Diseases), and the fish facility support team at the Kavli Institute for Systems Neuroscience.

# Additional information

## Funding

| Funder | Grant reference number | Author |
|---|---|---|
| Deutsche Forschungsge-meinschaft | SPP 1926 | Andreas Möglich<br>Dagmar Wachten |
| Deutsche Forschungsge-meinschaft | SPP 1726 | Dagmar Wachten |
| Deutsche Forschungsge-meinschaft | TRR83/SFB | Dagmar Wachten |
| Deutsche Forschungsge-meinschaft | Germany's Excellence Strategy - EXC2151 - 390873048 | Florian I Schmidt<br>Dagmar Wachten |
| Deutsche Forschungsge-meinschaft | FOR2743 | Dagmar Wachten |
| Deutsche Forschungsge-meinschaft | Emmy Noether | Florian I Schmidt |
| Boehringer Ingelheim Fonds | PhD fellowship | Jan Niklas Hansen |
| Samarbeidsorganet Helse Midt-Norge | grant | Nathalie Jurisch-Yaksi |
| Deutsche Forschungsge-meinschaft | SFB894/TP-A22 | David U Mick |

The funders had no role in study design, data collection and interpretation, or the decision to submit the work for publication.

## Author contributions

Jan N Hansen, Conceptualization, Data curation, Software, Formal analysis, Funding acquisition, Validation, Investigation, Visualization, Methodology, Writing - original draft, Writing - review and editing; Fabian Kaiser, Conceptualization, Data curation, Formal analysis, Validation, Visualization, Methodology, Writing - review and editing; Christina Klausen, Data curation, Formal analysis, Writing - review and editing; Birthe Stüven, Data curation, Formal analysis, Visualization, Writing - review and editing; Raymond Chong, Data curation; Wolfgang Bönigk, Resources, Data curation, Methodology; David U Mick, Conceptualization, Resources, Investigation, Methodology, Writing - original draft, Writing - review and editing; Andreas Möglich, Conceptualization, Resources, Funding acquisition, Investigation, Methodology, Writing - original draft, Writing - review and editing; Nathalie Jurisch-Yaksi, Conceptualization, Data curation, Formal analysis, Funding acquisition, Investigation, Visualization, Methodology, Writing - original draft, Writing - review and editing; Florian I Schmidt, Conceptualization, Resources, Funding acquisition, Writing - review and editing; Dagmar Wachten, Conceptualization, Data curation, Formal analysis, Supervision, Funding acquisition, Validation, Investigation, Writing - original draft, Project administration, Writing - review and editing

## Author ORCIDs

Jan N Hansen (iD) https://orcid.org/0000-0002-0489-7535
David U Mick (iD) http://orcid.org/0000-0003-1427-9412
Nathalie Jurisch-Yaksi (iD) http://orcid.org/0000-0002-8767-6120
Florian I Schmidt (iD) https://orcid.org/0000-0002-9979-9769
Dagmar Wachten (iD) https://orcid.org/0000-0003-4800-6332

## Ethics

Animal experimentation: The animal facilities and maintenance of the zebrafish, *Danio rerio*, were approved by the Norwegian Food Safety Authority (19/175222).

Decision letter and Author response
Decision letter https://doi.org/10.7554/eLife.57907.sa1
Author response https://doi.org/10.7554/eLife.57907.sa2

## Additional files

### Supplementary files
• Supplementary file 1. Plasmids and cloning information. Plasmids are listed and the ID and sequence of the primers that have been used for cloning are indicated.

• Transparent reporting form

### Data availability
All data generated or analysed during this study are included in the manuscript and are available through the following link: https://doi.org/10.6084/m9.figshare.c.4792248.

The following dataset was generated:

| Author(s) | Year | Dataset title | Dataset URL | Database and Identifier |
|---|---|---|---|---|
| Hansen JN, Kaiser F, Klausen C, Stüven B, Chong R, Bönigk W, Mick DU, Möglich A, Jurisch-Yaksi N, Schmidt FI, Wachten D | 2020 | Nanobody-directed targeting of optogenetic tools to study signaling in the primary cilium | https://doi.org/10.6084/m9.figshare.c.4792248 | figshare, 10.6084/m9.figshare.c.4792248 |

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
