## [Decision Letter]

**Acceptance summary:**

Cilia are critical signaling organelles. Tools are needed to delineate signaling events in the cilium that are distinct from those in the cell body. This work combines optogenetic and nanobody tools to study cAMP signaling in the primary cilium. The technique described here has the potential to be useful in studying additional ciliary signaling events.

**Decision letter after peer review:**

Thank you for submitting your article "Nanobody-directed targeting of optogenetic tools to study signaling in the primary cilium" for consideration by *eLife*. Your article has been reviewed by three peer reviewers, one of whom is a member of our Board of Reviewing Editors, and the evaluation has been overseen by Piali Sengupta as the Senior Editor. The following individual involved in review of your submission has agreed to reveal their identity: Kirk Mykytyn (Reviewer #3).

The reviewers have discussed the reviews with one another and the Reviewing Editor has drafted this decision to help you prepare a revised submission.

Summary:

The authors developed new techniques for studying ciliary signaling. Specifically, the authors combined optogenetic and nanobody tools to study cAMP signaling in a cilium. The enzymes that regulate cAMP levels were targeted to cilia and activated subsequently. They demonstrate that cAMP increase in cilia increase ciliary length whereas that in the cell body promotes ciliary shortening. The manuscript is well written, interesting and the tools developed may be useful for studying ciliary signaling in general. We have the following suggestions to improve the manuscript.

Essential revisions:

1) With regard to optogenetically manipulating ciliary cAMP levels, a limitation of the current study is the authors do not provide a direct physiological impact of increased ciliary cAMP levels. Although they demonstrate that cells possessing ciliary targeted bPAC have longer cilia after 16 hours of light stimulation, the assay did not activate bPAC specifically in the cilium. Rather, the entire cell was exposed to light, which will activate bPAC that is localized in the cell body. The authors should acknowledge that ciliary targeted constructs are not exclusively ciliary given the size limitation (Introduction, last paragraph, exclusively should be replaced with enriched). The study would be greatly strengthened if the authors could demonstrate a more immediate readout of optogenetic stimulation of cAMP specifically in the cilium. In the absence of this experiment (although addition of these data are recommended), you will need to alter the text to address this issue.

2) The data showed that increasing cAMP signaling in cilia promotes ciliary elongation whereas this increase in the cell body induces ciliary shortening. This is an interesting observation. However, in the fourth paragraph of the Discussion, the statement that an increase in intracellular cAMP levels contributes to an increase in cilia length seems to contradict the findings in this paper. This should be clarified. Forskolin is reported to stimulate ciliary length (Besschetnova et al., 2010), which is inconsistent with this work. The author should provide some explanation or speculation.

3) Figure 1E, if the values are calculated as mCherry/eGFP and eGFP fluorescence is decreasing then shouldn't the values increase after IBMX or forskolin treatment? Also, it would be helpful if the authors explained why different modulators of cAMP signaling were chosen for different assays.

4) Figure 4H, are the cilia in the before and after panels the same cilium? If not, why not? Similarly, Figure 3H, it would be helpful if videos or images of a time course were included to illustrate the changes in fluorescence.

5) Throughout the figures, it would be helpful if it was always indicated in the labels that the constructs were fused to a fluorophore. For example, in Figure 1D-G, indicate that bPAC and LAPD were fused to mCherry. Similarly, in Figure 2—figure supplement 1A legend, was mNphp3(201)-VHHlam-4 fused to eGFP? In Figure 1F legend, replace mNphp3(201)-bPAC-mCherry with mNphp3(201)-LAPD-mCherry. In Figure 2—figure supplement 2D legend, remove "cilia-targeted".

6) In the Discussion, there should be additional discussion about how versatile this nanobody-optogenetics method would be and also a discussion of possible limitations.

Specifically:

a) A large number of nanobodies are available, and a new nanobody can be developed if necessary. Therefore, we have a large number of choices for a tag that will be added to POI (protein of interest).

b) Addition of a tag to POI may interfere with its activity. Choice of a tag should be carefully considered so that addition of a tag or binding of nanobody to a tag will not influence activity of the POI.

c) Most importantly, what other ciliary signaling pathways can be studied by this method? For example, calcium signaling can be studied by this method but not Wnt or Shh signaling at this time.

d) what tags are currently used for ciliary targeting.

e) With regard to measuring cAMP levels in primary cilia, a limitation of this study is the authors have not demonstrated whether these techniques are sensitive enough to measure endogenous ciliary signaling. In Figure 4I, treatment of IMCD cells with 250 µM IBMX only results in a change in fluorescence of 0.1. It is unclear whether this readout is robust enough to detect changes in cAMP levels due to endogenous signaling, such as through a ciliary GPCR.

---

## [Author Response]

Essential revisions:1) With regard to optogenetically manipulating ciliary cAMP levels, a limitation of the current study is the authors do not provide a direct physiological impact of increased ciliary cAMP levels. Although they demonstrate that cells possessing ciliary targeted bPAC have longer cilia after 16 hours of light stimulation, the assay did not activate bPAC specifically in the cilium. Rather, the entire cell was exposed to light, which will activate bPAC that is localized in the cell body. The authors should acknowledge that ciliary targeted constructs are not exclusively ciliary given the size limitation (Introduction, last paragraph, exclusively should be replaced with enriched). The study would be greatly strengthened if the authors could demonstrate a more immediate readout of optogenetic stimulation of cAMP specifically in the cilium. In the absence of this experiment (although addition of these data are recommended), you will need to alter the text to address this issue.

We agree with the reviewers that it would be great to present another direct and physiological read-out for the light-stimulated increase of cAMP in the cilium. In addition, it would be great if we could combine the ciliary targeting approach with specific illumination of cilia independent of the rest of the cell. Although it cannot be excluded that small amounts of non-ciliary bPAC are activated by light, our targeting approach provides a strong enrichment in the cilium vs. cell body, and the ciliary length measurements clearly demonstrate the spatial resolution of the current experimental set-up. To add another level of complexity and read-out for the specific manipulation of cAMP signaling in the cilium, a combination of our approach with ciliary proteome analysis (e.g. Mick et al., 2015) would be perfect. This would allow to set the stimulus, i.e. an increase of cAMP levels, and then determine the phosphorylation status of known ciliary localized proteins, i.e. the Gli transcription factors, or even the localization of proteins that shuttle in and out of the cilium in a cAMP-dependent manner. This, however, is beyond the scope of this manuscript, but of course we have changed the wording in the Introduction according to the reviewer’s suggestion.

2) The data showed that increasing cAMP signaling in cilia promotes ciliary elongation whereas this increase in the cell body induces ciliary shortening. This is an interesting observation. However, in the fourth paragraph of the Discussion, the statement that an increase in intracellular cAMP levels contributes to an increase in cilia length seems to contradict the findings in this paper. This should be clarified. Forskolin is reported to stimulate ciliary length (Besschetnova et al., 2010), which is inconsistent with this work. The author should provide some explanation or speculation.

We regret any confusion caused by the sentence in question: “In line with this finding, an increase in intracellular cAMP levels and downstream activation of PKA has been shown to contribute to an increase in cilia length by regulating anterograde IFT transport velocity.” Indeed, the Forskolin data is puzzling and needs some more clarification. Forskolin activates transmembrane adenylyl cyclases, which have been identified in both, the cilium and cell body. Thus, Forskolin evokes an increase of the cAMP concentration in the cilium and the cell body. Thereby, the respective effect of cAMP signaling on ciliary length control would either be compensated or, depending on the expression level of ACs in either the cilium or cell body, increase or decrease of the ciliary length would be favored. This explains why some reports demonstrate an increase (Beschetnova et al., 2010), while others see a decrease upon Forskolin stimulation (Porpora et al., 2018). We also performed stimulation with Forskolin and detected no significant change in ciliary length because the data was rather variable (Figure 6A). To clarify and address this issue, we have added a paragraph in the Discussion.

For clarification, we also changed the following sentence in the Discussion: “Thus, cilia length control only seems to be sensitive to an increase, but not to a decrease in cAMP levels, and the direction of the regulation (lengthening or shortening of cilia) appears to be dependent on the origin of the cAMP increase”. We changed the sentence as follows: “Thus, cilia length control seems to rely on a spatially segregated increase, but not decrease of cAMP levels. The origin of the cAMP increase appears to determine whether cilia become longer or shorter.”

3) Figure 1E, if the values are calculated as mCherry/eGFP and eGFP fluorescence is decreasing then shouldn't the values increase after IBMX or forskolin treatment? Also, it would be helpful if the authors explained why different modulators of cAMP signaling were chosen for different assays.

The reviewer might be referring to Figure 3E, which illustrates the ciliary cAMP dynamics measured using 5-HT_6_-mCherry-cADDis. The reviewer is correct – the labeling of the y axis was wrong and has now been changed to “ciliary eGFP/mCherry fluorescence”.

In general, to pharmacologically stimulate cAMP levels in cells, we prefer to use the water-soluble Forskolin analog NKH477, which does not require addition of DMSO as a control. We routinely use NKH477 in HEK293 cells. However, in the literature on cAMP signaling in cilia, in particular in primary cilia, Forskolin seems to be predominantly used, maybe for historical reasons. This is why we also used Forskolin for the ciliated cells, i.e. IMCD-3 cells. In addition, we included IBMX, which blocks phosphodiesterases, as an alternative pharmacological stimulus to increase cAMP levels.

4) Figure 4H, are the cilia in the before and after panels the same cilium? If not, why not? Similarly, Figure 3H, it would be helpful if videos or images of a time course were included to illustrate the changes in fluorescence.

Cilia shown in Figure 3H represent different cilia – one contains the 5-HT6-cADDis sensor and mCherry, the other the 5-HT6-cADDis sensor and bPAC-mCherry. Images are shown for the first frame of imaging. Due to the experimental set-up, bPAC-mCherry is directly activated when starting the experiment. Thus, it is not possible to show before and after images, but only cilia with or without bPAC-mCherry. However, we have analyzed 8-10 cilia per condition and demonstrated consistent and significant differences between both conditions, underlining that this method is suited to study an increase of the ciliary cAMP concentration by photo-activation of bPAC.

Cilia shown in 4H indeed represent the same cilium for the respective before/after condition; during the recording, cilia move and, thereby, appear in the image in different orientations at different time points. In the revised version, we have included a video for the data shown in Figure 4H (Video 1), and also show more individual time points in the supplements to demonstrate the time course (Figure 4—figure supplement 1H).

5) Throughout the figures, it would be helpful if it was always indicated in the labels that the constructs were fused to a fluorophore. For example, in Figure 1D-G, indicate that bPAC and LAPD were fused to mCherry. Similarly, in Figure 2—figure supplement 1A legend, was mNphp3(201)-VHHlam-4 fused to eGFP? In Figure 1F legend, replace mNphp3(201)-bPAC-mCherry with mNphp3(201)-LAPD-mCherry. In Figure 2—figure supplement 2D legend, remove "cilia-targeted".

We have changed the labeling in all figures and in the text.

6) In the Discussion, there should be additional discussion about how versatile this nanobody-optogenetics method would be and also a discussion of possible limitations.Specifically:a) A large number of nanobodies are available, and a new nanobody can be developed if necessary. Therefore, we have a large number of choices for a tag that will be added to POI (protein of interest).

This proof of concept study describes an approach to specifically target a broad selection of optogenetic tools or sensors derived from GFP or DsRed to primary cilia. To expand nanobody-based targeting to this or any other organelle, a few epitope tags with matching nanobodies can be employed, including the synthetic 13 amino acid ALFA tag (Goetzke et al., bioRxiv 2019), the C-tag derived from α-synuclein (de Genst et al., 2010), and the SPOT-tag derived from β-catenin (Traenkle et al., 2015). Targeting via these tags is not only orthogonal to EGFP or mCherry-based targeting, but is also expected to only exhibit a minimal effect on the tagged protein. This may be particularly advantageous if nanobody binding to a fluorescent reporter protein or direct fusion to a fluorescent protein alters its optical properties. This has to be carefully evaluated for each POI. Importantly, epitope tags are also compatible with the reverse strategy: endogenous ciliary proteins can be tagged by CRISPR/Cas9-based genome-editing techniques, allowing optogenetic tools or sensors, fused to the respective nanobodies, to be recruited to ciliary proteins expressed at endogenous levels, and perhaps even to specific subdomains of primary cilia. Along the same lines, nanobodies against endogenous cilia proteins, e.g. Arl13b may reduce the necessary genetic modifications for cilia targeting. Nanobody-based binding controlled by light or by small molecules will further expand the versatility of this approach (Yu et al., 2019, Gil et al., 2019, Farrants et al., 2020). To date, all intracellular applications of nanobodies require genetic modification of target cells, a limitation that may be overcome with improved strategies for intracellular protein delivery. We have added another paragraph about the different choices for a tag etc. in the Discussion.

b) Addition of a tag to POI may interfere with its activity. Choice of a tag should be carefully considered so that addition of a tag or binding of nanobody to a tag will not influence activity of the POI.

See answer to a). We have included a paragraph addressing these points in the Discussion.

c) Most importantly, what other ciliary signaling pathways can be studied by this method? For example, calcium signaling can be studied by this method but not Wnt or Shh signaling at this time.

Our method is highly versatile, as it allows to both manipulate and measure cAMP signaling. Thus, ciliary signaling cascades involving cAMP would be a prime subject to investigate. In fact, this includes Shh signaling as stimulation with Shh has been proposed to reduce ciliary cAMP levels (Moore et al., PNAS 2018). Furthermore, GPCR signaling pathways use cAMP as a second messenger for signal transduction. Primary cilia are enriched in specific GPCRs, e.g. 5-HT6, V2R, EP4, Sstr3, or GPR120, just to name a few. Thus, the function of these GPCR-dependent signaling pathways could be analyzed with spatial and temporal resolution using our approach. Most importantly, the nanobody-based ciliary targeting applies to all proteins of interest that can be fused to a tag that is recognized by a nanobody. The tool box can be extended to analyze diverse signaling pathways, e.g. by targeting Ca^2+^, PKA activity, or lipid sensors, enzymes, or transcription factors to the primary cilium. We have amended the discussion accordingly.

d) what tags are currently used for ciliary targeting.

Common strategies involve direct fusion to the C terminus of either a full-length GPCR enriched in cilia, e.g. Sstr3 or 5-HT6, to a ciliary protein, e.g. Arl13b, or a truncated ciliary protein, e.g. the first 201 amino acids (aa) of the ciliary mouse Nphp3 protein. We had briefly mentioned these tags for ciliary targeting in the Introduction and have now extended this paragraph.

e) With regard to measuring cAMP levels in primary cilia, a limitation of this study is the authors have not demonstrated whether these techniques are sensitive enough to measure endogenous ciliary signaling. In Figure 4I, treatment of IMCD cells with 250 µM IBMX only results in a change in fluorescence of 0.1. It is unclear whether this readout is robust enough to detect changes in cAMP levels due to endogenous signaling, such as through a ciliary GPCR.

The reviewer is correct, we have not used a physiological stimulus, e.g. a ligand for a GPCR in mIMCD-3 cells. The main reason is that the knowledge about GPCRs, coupling to cAMP and for which an agonist has been identified is limited for mIMCD-3 cells, rendering it difficult to set a physiological stimulus. To circumvent this problem, other reports have fused the cAMP sensor to a GPCR, thereby ectopically expressing the GPCR and the biosensor, and then stimulated the cells with the respective ligand of the GPCR (e.g. Moore et al., PNAS 2018, Jiang et al., 2019). However, this approach includes the over-expression of a full-length GPCR, which might already alter ciliary cAMP signaling. We note that in sperm flagella, the mlCNBD-FRET biosensor was perfectly suited to measure changes of cAMP signaling in the physiological range (Mukherjee et al., 2016). Thus, we assume that also in primary cilia, physiological changes of the cAMP concentration can be reliably measured.